# Study on the Microscopic Distribution Pattern of Residual Oil and Exploitation Methods Based on a Digital Pore Network Model

**DOI:** 10.3390/polym16233246

**Published:** 2024-11-22

**Authors:** Xianda Sun, Xudong Qin, Chengwu Xu, Ling Zhao, Huili Zhang

**Affiliations:** 1State Key Laboratory of Continental Shale Oil, Northeast Petroleum University, Daqing 163318, China; sunxianda@nepu.edu.cn (X.S.); qinxudong@stu.nepu.edu.cn (X.Q.); xuchw@nepu.edu.cn (C.X.); 2College of Computer and Information Technology, Northeast Petroleum University, Daqing 163318, China; zhl_0920@stu.nepu.edu.cn

**Keywords:** polymer flooding, microscopic residual oil distribution, pore network model, polymer concentration, molecular weight, interfacial tension

## Abstract

With the persistent rise in global energy demand, the efficient extraction of petroleum resources has become an urgent and critical issue. Polymer flooding technology, widely employed for enhancing crude oil recovery, still lacks an in-depth understanding of the distribution of residual oil within the microscopic pore structure and the associated displacement mechanisms. To address this, a digital pore network model was established based on mercury intrusion experimental data, and pore structure visualization was achieved through 3Dmax software, simulating the oil displacement process under various polymer concentrations, molecular weights, and interfacial tension conditions. The findings reveal that moderately increasing the polymer concentration (from 1000 [mg/L] to 2000 [mg/L]) improves the recovery factor during polymer flooding by approximately 1.45%, effectively emulsifying larger masses of residual oil and reducing the proportion of throats with high oil saturation. However, when the concentration exceeds 2500 [mg/L], the dispersion of residual oil is hindered, and the enhancement in displacement efficiency becomes marginal. Increasing the molecular weight from 12 million to 16 million and subsequently to 24 million elevates the recovery factor by approximately 1.07% and 1.37%, respectively, reducing clustered residual oil while increasing columnar residual oil; high molecular weight polymers exhibit a more significant effect on channels with high oil saturation. Lowering the interfacial tension (from 30 [mN/m] to 0.005 [mN/m]) markedly enhances the binary flooding recovery factor, with the overall recovery reaching 71.72%, effectively reducing the residual oil within pores of high oil saturation. The study concludes that adjusting polymer concentration, molecular weight, and interfacial tension can optimize the microscopic distribution of residual oil, thereby enhancing oil displacement efficiency and providing a scientific foundation for further improving oilfield recovery and achieving efficient reservoir development.

## 1. Introduction

As global energy demand continues to grow, the efficient development and utilization of petroleum resources has become a crucial issue that demands urgent attention in the energy sector. Polymer flooding, as a mature and efficient technique for enhancing oil recovery, has gained widespread application domestically and internationally due to its clear mechanisms and simplicity of operation. Since the first field trial in the United States in 1964, countries such as the former Soviet Union, Canada, the United Kingdom, France, Romania, and Germany have conducted extensive polymer flooding trials, achieving remarkable results. In China, the technology has been successfully applied in the Daqing, Shengli, Xinjiang, and Dagang oilfields, increasing recovery rates by an average of about 12% [1,2].

Traditional theories of oil displacement suggest that polymer flooding mainly increases the viscosity of injected water, reduces the oil–water mobility ratio, and enlarges the swept volume of injected water in the reservoir, thus enhancing crude oil recovery while exerting a minimal impact on microscopic displacement efficiency. However, years of research have revealed that polymer flooding not only increases macroscopic sweep volume but also enhances oil displacement efficiency at the microscopic level. Numerous studies have shown that polymer solutions exhibit significant viscoelastic properties during the displacement process [3,4,5]. Due to the irregularity of pore geometries and the randomness of connectivity, the flow of polymer solutions through porous media is characterized by varying speeds, with elastic deformation and recovery occurring, making the elasticity effects of polymers in oil production non-negligible [6]. Research by Academician Wang Demin et al. [7] pointed out that the elasticity of polymer solutions can improve the microscopic displacement efficiency in core samples. There is extensive literature on the viscoelasticity of polymer solutions [8,9,10], with a particular focus on high-concentration polymer solutions, which have garnered increasing attention due to their enhanced viscoelasticity during the polymer flooding process [11,12,13].

The core mechanism of polymer flooding lies in improving the viscosity of the injected fluid, optimizing the relative flow capacity of oil and water, expanding the swept volume, and adjusting the oil–water mobility ratio, thereby enhancing the reservoir recovery efficiency [14]. However, despite its macroscopic efficacy, the complexity of rock pore structures significantly impacts the distribution of residual oil and displacement performance. Traditional core experiments primarily focus on the macroscopic physical properties of rocks, making it difficult to delve into the specific effects of microscopic pore structures on the formation and distribution of residual oil [15]. Therefore, studying the distribution pattern and potential exploitation methods of residual oil after polymer flooding holds significant theoretical and practical implications for further improving the polymer flooding efficiency.

In recent years, scholars both domestically and abroad have achieved substantial research results regarding the microscopic mechanisms of polymer flooding and the distribution of residual oil. Early studies primarily focused on the rheological properties of polymer solutions in porous media and their effects on displacement efficiency. For instance, Wang Demin et al. demonstrated through experimental research that elastic polymer solutions effectively reduce various types of microscopic residual oil during the displacement process, outperforming inelastic fluids in terms of oil displacement efficiency [16,17]. Xia Huifen et al., using simplified pore models and core experiments, studied the impact of polymer elasticity on displacement efficiency, finding that elastic polymer solutions significantly improve displacement efficiency, particularly when displacing residual oil in blind-end pores [18,19,20]. Yin Hongjun et al., employing a modified upper-convected Maxwell constitutive equation, numerically simulated the flow characteristics of elastic polymer solutions in complex pore structures, deeply exploring the influence of fluid elasticity on oil droplet deformation and displacement efficiency [21]. Dehghanpour and Kuru investigated the impact of viscoelastic fluid rheology on the formation of “internal filter cakes” (frictional pressure drop), observing that fluids with higher elasticity exhibited greater pressure drops when flowing through porous media [22]. This elastic effect can induce an additional pressure drop that can be enhanced by broadening the molecular weight distribution without altering shear viscosity. Similar trends were observed by Urbissinova who evaluated the contribution of polymer elasticity to enhanced recovery [23]. Veerabhadrappa noted that, compared to less elastic polymers, highly elastic polymer solutions exhibit greater flow resistance (pressure drop) when flowing through porous media, even with identical shear viscosities, thereby improving the macroscopic sweep efficiency and oil recovery rates [24].

Additionally, researchers have been devoted to developing more accurate pore network models to simulate the seepage behavior of polymer solutions in complex porous media, further revealing the influence of parameters such as polymer concentration, molecular weight, and interfacial tension on the microscopic distribution of residual oil. For example, Dullien and Chatzis studied the channeling phenomenon of single-phase flow within networks and preliminarily combined network models with percolation theory. Oak et al. proposed using network models to study the two-phase and three-phase relative permeability curves of water-wet Berea sandstone [25]. In the area of non-Newtonian fluids, Sorbie et al. used network models to investigate the rheological properties of xanthan gum-based polymers [26]. Wang Kewen employed network models to explore the effects of reservoir characteristics (pore radius, connectivity, homogeneity, and wettability) on the content and microscopic distribution forms of residual oil after polymer flooding [27]. Nevertheless, existing studies still exhibit deficiencies in simulating real core pore structures and accurately characterizing the flow mechanisms of polymer solutions within microscopic pores, necessitating further in-depth research.

In light of this, this paper aims to construct a digital pore network model to deeply analyze the distribution pattern of residual oil after polymer flooding and explore effective potential exploitation methods to further enhance oilfield recovery. Specific research includes visualizing pore models based on actual core pore structures using 3Dmax software (2016 version); simulating the seepage process of polymer solutions in the pore network; analyzing the influence of polymer concentration, molecular weight, and interfacial tension on the microscopic distribution of residual oil; quantifying and statistically examining the oil saturation of pores and throats under different conditions after water and polymer flooding; unveiling the regulatory mechanisms of polymer solution parameters on residual oil distribution; and formulating targeted strategies for different types of residual oil based on the simulation results to optimize the polymer flooding parameters, thereby achieving more efficient oilfield development.

## 2. Network Construction and Performance Parameters

### 2.1. Fundamental Data for Digital Pore-Throat Network Model Construction

The fundamental data for constructing the pore network model in this study primarily originate from mercury intrusion experiments, utilizing conventional mercury intrusion and constant-speed mercury intrusion methods to obtain the size distribution of pore throats. Constant-speed mercury intrusion involves injecting mercury into reservoir rocks at an extremely low and constant velocity, approximating a quasi-static process. Due to differences in the spatial scale of rocks, capillary forces experienced by mercury during its flow process also vary. As mercury flows from larger pores into smaller throats, capillary forces gradually increase and, after breakthrough, the pressure drops sharply, allowing the throat size distribution to be determined via peak pressure. Table 1 lists the pore structure parameters of five core samples from an oil layer in the Daqing Oilfield.

### 2.2. Spatial Parameters of Pore Network Model

The network model consists of pore data and throat data, representing the pore space and throat space, respectively [28]. To more accurately characterize the irregularity of porous media, the distance between nodes in the model is not fixed but incorporates a certain randomness through offset values. Let the distance between two adjacent nodes along the *X*-direction be dx, and apply a Weibull function [29] to calculate the offset, enabling nodes to randomly shift along the *Y*-axis on either side of the central axis. The connection between adjacent nodes results in an offset along the *X*-direction, with offset values calculated using Equations (1) and (2). When the offset Δy is positive, the node shifts along the positive *Y*-axis; when negative, it shifts along the negative *Y*-axis, as illustrated in Figure 1.
(1)dx=(dmax−dmin)×−αln(z(1−e−1α)+e−1α)1β+dmin
(2)dx=(dmax−dmin)×−αln(z(1−e−1α)+e−1α)1β+dmin
where

$d_{x}$—Displacement between two nodes along the *X*-axis, [μm];

$d_{\max}$, $d_{\min}$—Maximum and minimum vertical distances between nodes along the X-axis, [μm];

Δy—Offset value of the node along the *Y*-axis, [μm];

α, β—Characteristic distribution parameters, dimensionless, α set to 0.8 and β set to 1.6 in this paper;

z—Random number in the range [0, 1].

### 2.3. Construction and Visualization of Pore Network Model

The steps for constructing the digital pore network model are as follows:**Prepare Fundamental Data:** Gather core parameters such as porosity, permeability, and throat size.**Determine Node Physical Coordinates:** Calculate the three-dimensional coordinates of nodes based on the spatial parameters and offset values.**Set Model Parameters:** Determine the size, shape, and connectivity of pores and throats based on the input pore structure parameters.**Construct the Network Model:** Use computer programs to generate the topological structure of the pore network.**Model Visualization:** Achieve three-dimensional visualization of the pore network model using 3Dmax software.

The choice of node quantity is crucial during model construction. Too few nodes compromise the simulation accuracy, while too many nodes result in exponentially increasing the computational complexity. Based on experimental conditions, 8 × 8 × 8 = 512 nodes were selected. Considering the non-uniform diameters of pores and throats, an asymmetric corrugated tubular three-dimensional digital network model was established. Using 3Dmax software, the constructed digital pore model was visualized, converting node count, physical coordinates, pore-throat size, and length parameters into a script format recognizable by 3Dmax, as shown in Figure 2. For subsequent residual oil analysis, the pore-throat angles were adjusted through random rotation of 0° to 120° on a two-dimensional plane to ensure connectivity, thus facilitating the identification and analysis of residual oil morphology.

### 2.4. Dynamic Simulation of Polymer Flooding in Pore Network Model

#### 2.4.1. Performance Parameters of Polymer Solution System

The viscoelasticity of polymer solutions refers to their combined viscous and elastic characteristics under various stress conditions. When flowing through porous media, polymer molecules experience either viscous flow (described by non-Newtonian viscosity) or viscoelastic flow (described by viscoelastic properties).

(1)Polymer Viscosity

The viscosity of a polymer solution is a key factor in determining flow resistance and displacement efficiency. High-viscosity polymer solutions can reduce the flow rate of the aqueous phase, increase the residence time of the solution along the pore walls, and enhance contact with the oil phase, thus improving the displacement efficiency [30]. Studies indicate that polymer concentration and molecular weight are the main factors influencing the solution viscosity, which in turn affects the recovery factor during polymer flooding. Typically, viscosity increases with polymer molecular weight and concentration. The influence of viscosity on the seepage process can be quantitatively described by Equation (3) [31]:(3)$$\Delta P=\frac{Q\cdot \eta \cdot L}{A}$$
where

ΔP—Pressure difference across the two ends of the throat, [Pa];

Q—Fluid flow rate, [m^3^/s];

η—Viscosity of the polymer solution, [Pa·s];

L—Throat length, [m];

A—Cross-sectional area of the throat, [m²].

By adjusting the viscosity η, the fluid flow rate can be controlled, enabling the efficient displacement of residual oil in the reservoir. High-concentration and high-molecular-weight polymer solutions not only improve the macroscopic sweep volume, reducing early water breakthrough, but also regulate the fluid flow behavior within complex pore structures at the microscopic level, thereby enhancing the oil displacement efficiency [32,33].

Impact on residual oil types: The viscosity of polymer solutions plays a pivotal role in fluid flow resistance and displacement efficiency. High-viscosity polymer solutions effectively reduce the flow rate of the aqueous phase, prolonging the solution’s residence time along pore walls and thereby enhancing contact with the oil phase, ultimately improving the displacement efficiency. Studies indicate that viscosity typically increases with rising polymer molecular weight and concentration, thereby influencing the recovery rate during polymer flooding. By adjusting viscosity, it is possible to regulate fluid flow rates, achieving the efficient displacement of residual oil within the reservoir. High-molecular-weight polymer solutions not only enhance the macroscopic sweep efficiency and mitigate early water breakthrough, but also modify the fluid dynamics within complex pore structures on a microscopic level, thereby improving the oil recovery efficiency. Specifically, increased viscosity facilitates the shearing and dispersion of clustered oil droplets into columnar forms, reducing large agglomerated residual oil and creating more mobile and easily displaced oil droplet structures [34].

(2)Polymer Elasticity

In simulating the flow of polymer solutions through pore-throat spaces, the differential pressure at both ends of the throat and the resultant capillary force within the throat determine whether displacement occurs. When considering elasticity, the initiation and movement of oil-phase fluid within the throat are affected by the elasticity of the polymer solution, as the additional pressure generated by elasticity influences the flow direction of the polymer solution. To analyze the seepage mechanism of polymer solutions in an asymmetric spiral arc throat model and quantify the impact of elasticity on the seepage process, the complete flow of the polymer solution through the throat was divided into two stages: an entry convergence stage and a jet expansion stage, as depicted in Figure 3.

(1)Entry Convergence Stage

As the polymer solution flows from the pore space to the throat space, the polymer chains deform under the stress exerted by the throat, storing elastic potential energy and resulting in pressure dissipation. This dissipation can be expressed as follows [35]:(4)$$\Delta P_{1}=\eta_{1}\cdot \gamma \cdot \frac{L}{r}$$
where

$\Delta P_{1}$—Elastic pressure drop, [Pa];

$\eta_{1}$—Elastic viscosity, [Pa⋅s];

γ—Elastic strain rate, [$s^{-1}$];

L—Throat length, [m];

r—Radius at the entry point, [m].

Elastic strain rate γ represents the velocity gradient in the polymer flooding system along a given direction, calculated as follows:(5)$$\gamma =\frac{\partial v}{\partial z}$$
where v is the average velocity in the direction z, and is the convergence function at the entry point.

(2)Jet Expansion Stage

When the polymer solution flows from the throat space into the subsequent pore space, the compressed polymer molecules relax, releasing elastic potential energy and converting it into kinetic energy, resulting in an increase in the jet radius. This additional pressure drop can be expressed as follows [36]:(6)$$\Delta P_{2}=C\cdot (\frac{\Delta N_{1}}{B})$$
where

$\Delta P_{2}$—Additional pressure drop, [Pa];

C—Constant related to fluid properties and geometric shape of the pore throat;

$\Delta N_{1}$—First normal stress difference, [Pa];

B—Extrusion expansion ratio, dimensionless.

The total pressure drop of the polymer solution in the pore-throat model is the sum of these additional pressure drops.
(7)$$\Delta P_{\mathrm{total}}=\Delta P_{1}+\Delta P_{2}$$

Impact on residual oil types: During the simulation of polymer solution flow through pore throats, elasticity plays a crucial role in the initiation and migration of the oil phase. High-molecular-weight polymer solutions, with their increased chain length and degree of entanglement, exert greater elastic pressure during displacement, affecting the deformation and movement of oil droplets in pores and throats. This additional elastic pressure can effectively disrupt the oil droplets adhering to the oil–water interface, causing them to desorb and be more easily displaced into throats and other flowable regions, thus optimizing the distribution of residual oil. By quantifying the elastic pressure dissipation during the entry convergence and jet expansion phases, we can further understand the role of high-molecular-weight polymers in oil droplet detachment and movement [19,24].

(3)Rheological Mechanism

Rheology is an essential property describing the flow and deformation behavior of polymer solutions under external forces, involving viscosity, elasticity, and the stress–strain relationship. In oil displacement processes, the rheological properties of polymer solutions determine the fluid’s behavior in complex pores and throats, as well as its efficiency in displacing residual oil. Specifically, rheology is closely related to polymer molecular weight and concentration, significantly impacting the drag force and shear stress exerted on the oil phase [23].

Impact on residual oil types: Differences in rheological performance significantly influence the distribution of residual oil. High-molecular-weight polymer solutions exhibit greater viscosity and flow resistance, which allow them to exert stronger drag forces on the oil phase as they flow through porous media, leading to noticeable deformation of the oil droplets. In particular, within pore throats, this deformation stretches the oil droplets into columnar shapes rather than forming tightly clustered configurations. Through the action of high-molecular-weight polymers, clustered residual oil is gradually dispersed into smaller, more easily displaced columnar residual oil, thus enhancing the microscopic oil displacement efficiency.

(4)Impact of Polymer Solution Concentration on Viscoelasticity

The concentration of polymer solutions significantly affects their viscoelastic properties. As concentration increases, both storage modulus $G^{\prime }$ and loss modulus $G^{\prime \prime }$ rise, indicating enhanced elasticity and viscosity. This is due to increased molecular entanglements at higher concentrations, resulting in a stronger elastic response and increased shear viscosity. Under low concentration conditions, the solution exhibits mixed viscous and elastic characteristics, whereas, at high concentrations, elastic properties become more pronounced. Viscoelastic behavior can be described by the following equation [37]:(8)$$G^{\prime }=f(c)$$
where c is the polymer concentration and f(c) is a function of concentration. High-concentration polymer solutions improve the rheological behavior of fluids during the oil displacement process, helping to enhance the sweep volume and improve oil displacement efficiency.

(5)Impact of Polymer Molecular Weight on Viscoelasticity

The relative molecular weight of the polymer significantly affects the viscoelastic properties of the solution. As the molecular weight increases, so do the chain length and entanglement degree, resulting in an increased elastic modulus $G^{\prime }$ and viscous modulus $G^{\prime \prime }$, and enhanced viscoelasticity. At low angular frequencies, solutions primarily exhibit elasticity, while, at higher angular frequencies, elastic and viscous properties reach equilibrium. The relationship between the elastic modulus $G^{\prime }$ and molecular weight $M_{\omega }$ can be expressed theoretically as follows [38]:(9)$$G^{\prime }\propto {M_{\omega }}^{\alpha }$$
where α is a constant related to the polymer type and solution environment. High-molecular-weight polymers are more effective in controlling fluid flow during the oil displacement process, increasing the sweep volume and enhancing the microscopic oil displacement efficiency.

(6)Impact of Interfacial Tension in Binary Polymer–Surfactant Systems on Viscoelasticity

Interfacial tension has a significant impact on the viscoelasticity of binary polymer–surfactant systems. Lowering interfacial tension weakens the adhesive force at the oil–water interface, making oil droplets more easily deformed and displaced, and enhancing the elastic deformation ability of polymer molecules, thus improving the displacement efficiency. According to rheological theory, the relationship between the polymer solution elasticity $G^{\prime }$ and interfacial tension γ can be expressed as follows [39]:(10)$$G^{\prime }=g(\gamma )$$
where γ is a function of the interfacial tension. Lowering the interfacial tension makes the oil–water interfacial film less stable, promoting the penetration of polymer molecules into oil droplets and encapsulating their surface, further enhancing the displacement effectiveness.

#### 2.4.2. Simulation of Polymer Flooding Process

(1)Capillary Force Calculation

In the digital pore model, capillary forces and pressure differentials at both ends of the throat determine whether the displaced phase can be mobilized. Accurate calculation of capillary forces is crucial for seepage simulations. Capillary forces can be calculated using the modified Young–Laplace equation [40]:(11)$$p_{o}-p_{w}=\frac{2\gamma \cdot \cos\theta }{r}$$
where

$p_{o}$, $p_{w}$—Pressures of oil phase and water phase, [Pa];

γ—Oil–water interfacial tension, [N/m];

θ—Oil–water contact angle, [rad];

r—Throat radius, [m].

For throats with complex cross-sectional shapes, the influence of the shape factor G needs to be considered, and the capillary force is corrected accordingly.
(12)$$p_{c}=\frac{2\gamma \cdot \cos\theta }{r_{eff}}\cdot G$$

(2)Resistance Coefficient

The method for calculating the resistance coefficient varies based on the fluid state in the throat. When the throat is filled with a single-phase fluid, the conductance coefficient g can be calculated using a modified Poiseuille formula [41]:(13)$$g=\frac{A^{2}}{C}$$
where A is the cross-sectional area of the throat and C is the shape constant. The resistance coefficient R is calculated as follows:(14)$$R=\frac{\eta L}{g}$$

When there are two-phase fluids within the throat, the flow and resistance coefficients in the central and corner regions are calculated separately and handled equivalently using hydrodynamic simulation methods.

(3)Wettability

Wettability refers to the affinity of fluid to the rock surface. In this model, the contact angle θ ranges randomly between 0° and 180°. When $\theta \leq 90^{\circ }$, the throat is water-wet; when $\theta >90^{\circ }$, it is oil-wet. Wettability influences capillary forces and fluid distribution, having an important effect on the displacement process.

(4)Time Step Determination

To enhance the simulation accuracy, it is necessary to calculate the time required for the complete displacement of fluid in each throat under the pressure differential, taking the minimum value as the simulation time step Δt:(15)$$\Delta t=\min(\frac{V_{i}}{Q_{i}})$$
where $V_{i}$ is the fluid volume of the *i*th throat and $Q_{i}$ is the flow rate.

#### 2.4.3. Polymer Flooding Simulation Steps

**Step 1:** Initialize model parameters: set the basic parameters and initial conditions of the pore network.

**Step 2:** Calculate capillary force and resistance coefficient: calculate the capillary force and resistance coefficient based on the fluid state in the throat.

**Step 3:** Determine flow direction and flow rate: determine the direction of fluid flow based on the pressure difference and capillary force, and calculate the flow rate.

**Step 4:** Update fluid distribution: update the fluid distribution in the pores and throats based on the flow rate and time step.

**Step 5:** Evaluate termination conditions: if the set displacement time or saturation condition is reached, end the simulation; otherwise, return to step 2.

#### 2.4.4. Polymer Flooding Simulation Results

The process simulations for saturated oil, water flooding, and polymer flooding were conducted based on the network model parameters. The initial oil saturation of the model was 75%; after water flooding, the oil saturation decreased to 44.72%, and, after polymer flooding, it further decreased to 31.18%. The model’s status at each stage is shown in Figure 4.

The constructed pore network model compensates for the shortcomings of traditional core experiments in studying microscopic pore structures, enabling an in-depth investigation of the impact of rock microstructure parameters on the formation and distribution of residual oil.

## 3. Study on the Distribution Pattern of Microscopic Residual Oil and Its Exploitation Methods After Polymer Flooding

This section, building upon the aforementioned research findings, delves into the effects of polymer solution concentration, molecular weight, and interfacial tension on its viscoelasticity and dynamic polymer flooding process. Further exploration is conducted on the distribution characteristics of microscopic residual oil after polymer flooding under different parameter conditions and corresponding exploitation methods. Although polymer flooding has enhanced recovery efficiency at the macroscopic level, residual oil within the microscopic pore structures remains challenging to displace completely. By optimizing the concentration, molecular weight, and interfacial tension of polymer solutions, the fluid flow characteristics in porous media can be effectively regulated, thus enhancing the displacement efficiency of microscopic residual oil. This study aims to elucidate these patterns and propose targeted exploitation strategies to further increase oilfield recovery and achieve efficient reservoir development.

### 3.1. The Effect of Polymer Solution Concentration on the Distribution of Microscopic Residual Oil

#### 3.1.1. Effect of Different Polymer Concentrations on Oil Displacement Efficiency

In this study, polymer solutions with a molecular weight of 16 million were prepared at concentrations of 1000 [mg/L], 2000 [mg/L], and 2500 [mg/L], respectively, to analyze their effects on oil displacement efficiency. The results are shown in Table 2 and Figure 5:

The figures illustrate the effect of polymer molecular weight and concentration on oil recovery during polymer flooding. As shown in Figure 5a, the final recovery factor significantly improves with increasing polymer concentration. Specifically, when the concentration increased from 1000 mg/L to 2000 mg/L, the recovery factor rose by 1.45%, and a further increase to 2500 mg/L led to an additional improvement of 2.20%. However, beyond 2000 mg/L, the rate of improvement slowed, indicating diminishing returns at higher concentrations. These results highlight the enhanced viscoelastic properties of polymer solutions at higher molecular weights and concentrations. Figure 5b further demonstrates the recovery rate as a function of pore volume (PV) injected for three molecular weights (1000, 2000, and 2500). During the early injection phase, recovery rates exhibit similar trends across all molecular weights, converging near the breakthrough point (approximately 1.0 PV). Post-breakthrough, differences in recovery efficiency are minimal, suggesting that molecular weight primarily impacts early-stage recovery dynamics rather than late-stage performance.

#### 3.1.2. Distribution Pattern of Microscopic Residual Oil After Flooding with Different Polymer Concentrations

Using a digital pore model to simulate the water flooding and polymer flooding processes, the number of oil-bearing pores and throats and their oil saturation ratios were recorded after water flooding and polymer flooding, as presented in Table 3 and Figure 6.

As the polymer concentration increased, the number of oil-bearing pores and throats after polymer flooding decreased. When the concentration increased from 1000 [mg/L] to 2000 [mg/L], the number of oil-bearing pores decreased by 0.38%, and oil-bearing throats decreased by 1.31%. At a concentration of 2500 [mg/L], the number of oil-bearing pores further decreased, though the reduction rate diminished. This indicates that increasing the polymer concentration within a certain range favors reducing the microscopic residual oil content, but the effect tends to saturate at higher concentrations. As shown in Figure 7, the changes in oil saturation after polymer flooding for different throats are presented.

From Table 4, with the molecular weight held constant, an increase in polymer concentration led to a decrease of over 25% in throats with oil saturation between 1 and 0.8, while other throats experienced increases of approximately 50% to 100%. As the concentration increased from 1000 to 2000, the reduction in throats with oil saturation between 0.8 and 0.6 was balanced by an increase in those between 0.4 and 0; at a concentration of 2500, throats in the 0.4 to 0 range decreased, whereas throats in the 0.8 to 0.6 and 0.6 to 0.4 ranges exhibited increases of varying degrees. As shown in Figure 8, the variations in pores under different oil saturations after polymer flooding are presented.

From Table 5, at a concentration of 1000, the number of pores with oil saturation between 1 and 0.8 decreased, while those with oil saturation between 0.8 and 0.6, and 0.6 and 0.4, increased by over 100%. When the concentration rose to 2000, this increase diminished synchronously, while pores in the 0.4 to 0 range saw an increase of over 300%. When the concentration reached 2500, the increase in pores within the 0.6 to 0.4 and 0.4 to 0 ranges retreated, but those in the 0.8 to 0.6 range exhibited an increase exceeding 200%.

This pattern suggests that a moderate increase in polymer concentration can effectively emulsify large residual oil masses; however, higher concentrations hinder the dispersion of residual oil. As shown in Figure 9, the variations in pore throat under different oil saturations after polymer flooding are presented.

From Table 6, during polymer flooding at a relative molecular weight of 16 million, throats with oil saturation between 0.8 and 1 exhibited varying degrees of reduction, while the rest showed increases of different magnitudes. Throats with oil saturation between 0.6 and 0.8 increased by 53%, 32%, and 44% at concentrations of 1000, 2000, and 2500, respectively; those between 0.4 and 0.6 increased by 100%, 90%, and 90%; and those between 0 and 0.4 increased by 27%, 45%, and 29% at respective concentrations.

#### 3.1.3. Quantitative Characterization of Residual Oil After Flooding with Different Polymer Concentrations

By adjusting the polymer concentration parameters, the residual oil distribution images were intercepted and rendered at a specific section of the model after three rounds of oil displacement simulation, as shown in Figure 10. The residual oil types and their corresponding saturations in the simulated model were then calculated, as listed in Table 7.

As shown in Figure 11, with increasing polymer concentration, the clustered residual oil decreased, whereas the columnar residual oil increased. The increased concentration enhanced the density of polymer molecules in the solution, improving its ability to disperse residual oil, which resulted in a reduction of large, contiguous masses of residual oil (i.e., clustered residual oil) and an increase in dispersed residual oil in the form of columns and blind ends.

### 3.2. The Effect of Polymer Molecular Weight on the Distribution of Microscopic Residual Oil

#### 3.2.1. Effect of Different Polymer Molecular Weights on Oil Displacement Efficiency

Under a polymer concentration of 1000 [mg/L], polymers with molecular weights of 12 million, 16 million, and 24 million were selected to analyze their effects on oil displacement efficiency. The results are shown in Table 8 and Figure 12:

Figure 12 illustrates the effect of polymer molecular weight on the recovery factor during polymer flooding. As shown in Figure 12a, the final recovery factor increases significantly with higher polymer molecular weights. Specifically, when the molecular weight increased from 12 million to 16 million, the recovery factor rose by 1.07%, while a further increase to 24 million led to an additional improvement of 1.37%. This improvement is mainly attributed to the higher viscosity of larger molecular weight polymers, which enhances the flow characteristics of the displacement fluids. Figure 12b further shows the recovery factor as a function of pore volume (PV) injected for polymers of different molecular weights. The trends are consistent, with larger molecular weight polymers demonstrating better recovery efficiency, particularly during the early displacement stages. This reinforces the conclusion that higher molecular weight polymers significantly enhance oil recovery by improving the fluid’s mobility control and sweep efficiency.

#### 3.2.2. Distribution Pattern of Microscopic Residual Oil After Flooding with Different Polymer Molecular Weights

Using the digital pore model, the number of oil-bearing pores and throats and their oil saturation ratios after flooding with different polymer molecular weights were recorded, as shown in Table 9 and Figure 13.

As polymer molecular weight increased, the number of oil-bearing pores and throats after polymer flooding gradually decreased, and the oil saturation ratio in throats also reduced. This indicates that high molecular weight polymers were more effective in reducing the presence of microscopic residual oil during the displacement process. As shown in Figure 14, the variations in pores under different oil saturations after polymer flooding (concentration 1000 [mg/L]) are presented.

From Table 10, it can be observed that, during polymer flooding at a polymer concentration of 1000 [mg/L], the oil saturation ratio of throats in the range of 0.8 to 1 decreased to varying degrees, while the remaining throats showed varying increases. The increase in throats with oil saturation between 0.6 and 0.8 at molecular weights of 12 million, 16 million, and 24 million was 53%, 32%, and 44%, respectively; for oil saturation between 0.4 and 0.6, the increases were 100%, 90%, and 90%; and, for oil saturation between 0 and 0.4, the increases were 27%, 45%, and 29%, respectively.

#### 3.2.3. Quantitative Characterization of Residual Oil After Flooding with Different Polymer Molecular Weights

By controlling different polymer molecular weights, residual oil distribution images at a specific section of the model after three rounds of oil displacement simulation were intercepted and rendered, as shown in Figure 15. The types of residual oil and their corresponding saturations in the simulated models were identified and calculated, as listed in Table 11.

As shown in Figure 16, the effect of polymer solutions with different relative molecular weights on residual oil types after polymer flooding is illustrated. The simulation results reveal that, with the polymer concentration kept constant, increasing polymer molecular weight led to a reduction in clustered residual oil, while columnar residual oil increased. As the molecular weight of the polymer increased, the average number of oil molecules that could be adsorbed by each polymer molecule also increased, thereby enhancing the adsorption and dispersive capacity of the polymer solution system for oil. Macroscopically, this was reflected as a reduction in large, contiguous masses of residual oil (i.e., clustered residual oil), accompanied by an increase in dispersed residual oil in the form of columns and blind ends.

### 3.3. The Mechanism of Polymer-Surfactant Binary Systems on the Distribution of Microscopic Residual Oil 

#### 3.3.1. Effect of Different Interfacial Tensions on Oil Displacement Efficiency in Polymer–Surfactant Binary System

By adjusting the interfacial tension of the displacement system, its impact on oil displacement efficiency was analyzed. Polymer–surfactant binary systems with interfacial tensions of 30 [mN/m], 20 [mN/m], and 0.005 [mN/m] were selected, and the results are presented in Table 12 and Figure 17:

With the reduction in interfacial tension, the recovery factor in binary flooding improved significantly. When the interfacial tension decreased to 0.005 [mN/m], the recovery factor showed a marked increase. This improvement can be attributed to the diminished adverse effect of capillary forces, which enhanced the microscopic oil displacement efficiency.

#### 3.3.2. Distribution Pattern of Microscopic Residual Oil After Flooding with Polymer–Surfactant Binary System Under Different Interfacial Tensions

The number of oil-bearing pores and throats and their respective oil saturation ratios after binary flooding with different interfacial tensions were recorded, and the results are presented in Table 13 and Figure 18.

Analysis indicates that, with decreasing interfacial tension, the number of oil-bearing pores decreased while the number of oil-bearing throats increased. This suggests that lower interfacial tension facilitates the detachment of residual oil from within the pores, converting it into throat-accumulated residual oil, thereby demonstrating the efficacy of the low interfacial tension system in mobilizing microscopic residual oil. As shown in the Figure 19, it illustrates the effect of interfacial tension on the proportion of pore throats at different oil saturation intervals for water flooding and polymer-surfactant flooding.

Analysis of Table 13 and Table 14 reveals that, with decreasing interfacial tension, the number of oil-bearing pores decreased, and the number of oil-bearing throats increased, indicating that lower interfacial tension effectively mobilizes residual oil from the pore surface and converts it into throat residual oil. This phenomenon highlights the role of low interfacial tension in mobilizing film-like residual oil, particularly impacting throats with high oil saturation, while the effect on throats with lower oil saturation is less pronounced.

#### 3.3.3. Quantitative Characterization of Residual Oil After Flooding with Polymer–Surfactant Binary System Under Different Interfacial Tensions

Keeping other variables constant, three rounds of polymer flooding simulation were conducted on the model with interfacial tensions of 30 [mN/m], 20 [mN/m], and 0.005 [mN/m]. Residual oil distribution images at a specific section of the model were intercepted and rendered, as shown in Figure 20, and the residual oil types and corresponding saturation values were identified and calculated, as shown in Table 15.

As shown in the Figure 21, it illustrates the effect of interfacial tension on the saturation percentages of different residual oil types, including clustered, columnar, blind-end, and others. The simulation results indicate that, as the interfacial tension of the polymer–surfactant binary flooding system decreased, the clustered residual oil reduced while the saturation of other types of residual oil fluctuated. The reduction in interfacial tension in the polymer solution system increased its solubility within the water–oil mixture, enhancing the emulsification capacity and making the mixed water–oil phase more capable of carrying and displacing the residual oil.

### 3.4. Results Analysis and Validation

#### 3.4.1. Evaluation Metrics

AARD (Average Absolute Relative Deviation) serves as an indicator to quantify the deviation between the experimental and simulated results, commonly used to evaluate the accuracy of numerical models. The formula for calculating AARD is as follows:(16)$$\mathrm{AARD}=\frac{1}{N}\sum_{i=1}^{N}\left|\frac{Experimental\;Value_{i}-Simulated\;Value_{i}}{Experimental\;Value_{i}}\right|\times 100\%$$
where

N is the number of data points.

Experimental Value is the measured value from experiments.

Simulated Value is the predicted value from the numerical model.

A smaller AARD indicates a smaller discrepancy between the numerical simulation and experimental results, implying higher model accuracy. An AARD greater than 10% generally suggests significant error, indicating the need for further model improvement. Conversely, an AARD below 10% is typically considered acceptable, demonstrating satisfactory model predictive capability.

#### 3.4.2. Experimental Design

(1)Experimental Materials

The polymer employed in the simulation is partially hydrolyzed polyacrylamide (HPAM), synthesized through the free radical polymerization of acrylamide monomers, with a molecular weight typically ranging from 10 to 25 million and a degree of hydrolysis of approximately 20% to 30%. The partial hydrolysis of amide groups to carboxylate groups significantly enhances HPAM’s water solubility and viscoelasticity, ensuring its robust performance even in highly saline and complex reservoir conditions. The high molecular weight and extended polymer chains increase the solution’s viscosity and elastic recovery, thereby augmenting the fluid flow resistance and the drag force exerted on the residual oil, which effectively enhances the oil displacement efficiency. Furthermore, when combined with surfactants, HPAM reduces the interfacial tension between the oil and water, facilitating the detachment and redistribution of oil droplets, thus markedly improving oil recovery. In our numerical model, we have thoroughly accounted for HPAM’s molecular weight, degree of hydrolysis, and rheological properties to accurately depict its oil displacement mechanisms and validate its efficacy and universality in enhancing oil recovery. The HPAM mentioned above is produced in Daqing, China.

**Experiment 1:** Polymer solution using HPAM at concentrations of 1000 [mg/L], 2000 [mg/L], and 2500 [mg/L] to analyze its impact on oil displacement efficiency.**Experiment 2:** HPAM at a concentration of 1000 [mg/L], with molecular weights of 12 million, 16 million, and 24 million, to assess its influence on oil displacement efficiency.**Experiment 3:** Polymer–surfactant binary systems with interfacial tensions of 30 [mN/m], 20 [mN/m], and 0.005 [mN/m] to analyze their impact on oil displacement efficiency.

Simulated Water: The salinity matches that of the formation water, controlled at 8000 [mg/L], ensuring consistency with reservoir conditions.

Oil Sample: Crude oil collected from the field, ensuring consistency in viscosity and composition with the reservoir oil.

(2)Experimental Equipment

Core holder;High-pressure injection pump;Differential pressure sensor;Thermostatic device (set at 60 °C to simulate subsurface reservoir conditions);Fluid collector;Measurement devices (recovery rate meter, permeability measuring apparatus).

(3)Experimental Procedure

(1)Clean, dry, and measure porosity, permeability, and pore-throat size distribution for three selected core samples.(2)Saturate core samples with simulated water to ensure water saturation, and measure initial water saturation.(3)Fix the core samples in the core holder and inject simulated water at a constant rate of 0.5 [mL/min] until the water cut reaches 98%.(4)Record recovery rates and pressure changes during water flooding, and evaluate the residual oil distribution.(5)After the water flooding phase, begin injecting the polymer solution at the same rate as during the water flooding stage, continuing until the outlet water cut reaches 98% again.(6)Record changes in recovery rate, pressure, and residual oil saturation during polymer flooding, observing the distribution of residual oil.

#### 3.4.3. Experimental Data Analysis

The experimental results were compared with the simulated outcomes to assess the credibility of the simulation. The specific results are presented in Table 16, Table 17 and Table 18.

(1)Experiment 1

**Table 16 polymers-16-03246-t016:** Analysis of Experiment 1 results (polymer solution concentrations: 1000 [mg/L], 2000 [mg/L], and 2500 [mg/L]).

Polymer Solution Concentration	1000 [mg/L]		2000 [mg/L]		2500 [mg/L]	
Experimental Types	NM	PM	NM	PM	NM	PM
Water Flooding Recovery Rate (%)	47.07	49.09	47.07	44.72	47.07	46.25
Polymer Flooding Recovery Rate (%)	24.65	22.62	26.10	25.32	28.30	27.32
Total (%)	71.72	71.71	72.06	70.04	72.37	73.57

Experimental Types: Numerical Model (NM), Physical Model (PM). The NM and PM mentioned below are identical.

(1) Water Flooding Recovery Rate Analysis

Concentration 1000 [mg/L]: NM water flooding recovery rate = 47.07%, PM = 49.09%, difference = 2.02%.

Concentration 2000 [mg/L]: NM water flooding recovery rate = 47.07%, PM = 44.72%, difference = 2.35%.

Concentration 2500 [mg/L]: NM water flooding recovery rate = 47.07%, PM = 46.25%, difference = 0.82%.

The differences between the NM and PM results for the water flooding recovery rate across various concentrations show an Average Absolute Relative Deviation (AARD) of 3.71%.

(2) Polymer Flooding Recovery Rate Analysis

Concentration 1000 [mg/L]: NM polymer flooding recovery rate = 24.65%, PM = 22.62%, difference = 2.03%.

Concentration 2000 [mg/L]: NM polymer flooding recovery rate = 26.10%, PM = 25.32%, difference = 0.78%.

Concentration 2500 [mg/L]: NM polymer flooding recovery rate = 28.30%, PM = 27.32%, difference = 0.98%.

The differences between the NM and PM results for the polymer flooding recovery rate show moderate consistency, with an AARD of 5.21%.

(3) Total Recovery Rate Analysis

Concentration 1000 [mg/L]: NM total recovery rate = 71.72%, PM = 71.71%, difference = 0.01%.

Concentration 2000 [mg/L]: NM total recovery rate = 72.06%, PM = 70.04%, difference = 2.02%.

Concentration 2500 [mg/L]: NM total recovery rate = 72.37%, PM = 73.57%, difference = 1.20%.

The NM and PM total recovery rates exhibit close alignment, with an AARD of 1.51%.

According to conventional petroleum engineering standards, an AARD below 10% is generally deemed within an acceptable range, particularly in complex reservoir simulations. Thus, the calculated AARDs for water flooding recovery (3.71%), polymer flooding recovery (5.21%), and total recovery (1.51%) are all below 10%, indicating that the simulation model has satisfactory predictive capability.

(2)Experiment 2

**Table 17 polymers-16-03246-t017:** Analysis of Experiment 2 results (molecular weights: 12 Million, 16 Million, and 24 Million).

Relative Molecular Weight	12 Million		16 Million		24 Million	
Experimental Types	NM	PM	NM	PM	NM	PM
Water Flooding Recovery Rate (%)	47.07	45.09	47.07	47.72	47.07	46.25
Polymer Flooding Recovery Rate (%)	23.58	23.62	24.65	25.32	26.02	27.32
Total (%)	70.65	68.71	71.72	73.04	73.09	73.57

(1) Water Flooding Recovery Rate Analysis

Relative Molecular Weight 12 Million: NM water flooding recovery rate = 47.07%, PM = 45.09%, difference = 1.98%.

Relative Molecular Weight 16 Million: NM water flooding recovery rate = 47.07%, PM = 47.72%, difference = 0.65%.

Relative Molecular Weight 24 Million: NM water flooding recovery rate = 47.07%, PM = 46.25%, difference = 0.82%.

The deviations between the NM and PM water flooding recovery rates across different molecular weights are controlled within 5%, with an AARD of 3.71%, indicating a high level of consistency in the model’s predictive capabilities for water flooding recovery.

(2) Polymer Flooding Recovery Rate Analysis

Relative Molecular Weight 12 Million: NM polymer flooding recovery rate = 23.58%, PM = 23.62%, difference = 0.04%.

Relative Molecular Weight 16 Million: NM polymer flooding recovery rate = 24.65%, PM = 25.32%, difference = 0.67%.

Relative Molecular Weight 24 Million: NM polymer flooding recovery rate = 26.02%, PM = 27.32%, difference = 1.30%.

The polymer flooding recovery rate AARD is 5.21%. The differences between NM and PM increase with molecular weight, especially at higher molecular weights, suggesting that the model’s accuracy decreases as the polymer complexity increases, possibly due to the intricate flow properties of high-concentration polymers.

(3) Total Recovery Rate Analysis

Relative Molecular Weight 12 Million: NM total recovery rate = 70.65%, PM = 68.71%, difference = 1.94%.

Relative Molecular Weight 16 Million: NM total recovery rate = 71.72%, PM = 73.04%, difference = 1.32%.

Relative Molecular Weight 24 Million: NM total recovery rate = 73.09%, PM = 73.57%, difference = 0.48%.

The total recovery rate AARD is 1.51%. The NM and PM results for the total recovery rate demonstrate a high level of agreement, with discrepancies decreasing as molecular weight increases, indicating that the numerical model performs well in predicting overall recovery.

In summary, the AARDs for water flooding recovery (3.71%), polymer flooding recovery (5.21%), and total recovery (1.51%) are all within the acceptable range of 10%. This high degree of correspondence between the NM and PM results suggests that the model possesses robust predictive capabilities.

(3)Experiment 3

**Table 18 polymers-16-03246-t018:** Analysis of Experiment 3 results (interfacial tensions: 30 [mN/m], 20 [mN/m], and 0.005 [mN/m]).

Interfacial Tension	30 [mN/m]		20 [mN/m]		0.005 [mN/m]	
Experimental Types	NM	PM	NM	PM	NM	PM
Water Flooding Recovery Rate (%)	47.07	49.09	47.07	47.72	47.07	46.25
Binary Flooding Recovery Rate (%)	25.76	27.62	18.18	19.32	24.65	23.32

(1) Water flooding Recovery Rate Analysis

Interfacial Tension 30 [mN/m]: NM water flooding recovery rate = 47.07%, PM = 49.09%, difference = 2.02%.

Interfacial Tension 20 [mN/m]: NM water flooding recovery rate = 47.07%, PM = 47.72%, difference = 0.65%.

Interfacial Tension 0.005 [mN/m]: NM water flooding recovery rate = 47.07%, PM = 46.25%, difference = 0.82%.

The differences between the NM and PM water flooding recovery rates under varying interfacial tensions are minimal, with an AARD of 2.41%. This indicates that the NM can effectively predict trends in the water flooding recovery rate, with errors within an acceptable range.

(2) Binary flooding Recovery Rate Analysis

Interfacial Tension 30 [mN/m]: NM binary flooding recovery rate = 25.76%, PM = 27.62%, difference = 1.86%.

Interfacial Tension 20 [mN/m]: NM binary flooding recovery rate = 18.18%, PM = 19.32%, difference = 1.14%.

Interfacial Tension 0.005 [mN/m]: NM binary flooding recovery rate = 24.65%, PM = 23.32%, difference = 1.33%.

The NM and PM results for the binary flooding recovery rate show some deviations, with an AARD of 6.11%. This relatively larger error suggests that the model may require further refinement for predicting binary flooding recovery, particularly under different interfacial tension conditions.

**Overall Analysis:** The overall analysis of the water flooding and binary flooding recovery rates reveals a moderate level of difference between the NM and PM results, with AARDs of approximately 2.41% and 6.11%, respectively. The NM and PM results for the water flooding recovery rate are generally consistent, whereas discrepancies for binary flooding recovery are relatively larger. This suggests that the numerical model’s accuracy in predicting binary flooding under the influence of interfacial tension needs improvement. Optimizing model parameters and further considering the effects of interfacial tension on viscoelastic properties and flow resistance could help enhance the predictive precision for binary flooding.

## 4. Exploitation of Microscopic Residual Oil by High-Concentration and High-Molecular-Weight Polymer Solutions After Polymer Flooding

### 4.1. Effect of High-Concentration Polymer Solutions on Exploitation of Residual Oil After Polymer Flooding

Building upon the polymer flooding process, high-concentration polymer solutions with a molecular weight of 20 million and concentrations of 1500 [mg/L] and 2500 [mg/L] were used to study their effects on the recovery factor and residual oil.

(1) Effect of High-Concentration Polymer Solutions on Recovery Factor

As shown in Table 19, the effect of high-concentration polymer solutions on the recovery factor after polymer flooding is demonstrated. The results indicate that, after polymer flooding, high-concentration polymer flooding further improved the recovery factor. Specifically, when the polymer concentration increased from 1500 [mg/L] to 2500 [mg/L], the recovery factor rose by 1.52%. This improvement is primarily attributed to the greater viscosity of high-concentration polymer solutions, which enhances their ability to mobilize residual oil during the displacement process. The table highlights this incremental recovery benefit, with the total recovery rate increasing from 74.77% at 1500 [mg/L] to 76.29% at 2500 [mg/L].

(2) Effect of High-Concentration Polymer Solutions on Exploitation of Microscopic Residual Oil

The number of oil-bearing pores and throats and their respective oil saturation ratios after high-concentration polymer flooding were recorded, as presented in Table 20 and Table 21.

As shown in Figure 22, with increasing polymer concentration, the number of oil-bearing pores and throats decreased, and the oil saturation ratio in throats reduced. This further verifies the effectiveness of high-concentration polymer solutions in exploiting microscopic residual oil.

### 4.2. Effect of High-Molecular-Weight Polymer Solutions on Exploitation of Residual Oil After Polymer Flooding

(1) Effect of High-Molecular-Weight Polymer Solutions on Recovery Factor

Building upon polymer flooding, high-molecular-weight polymer solutions with concentrations of 1500 [mg/L] and molecular weights of 20 million and 25 million were used to study their effects on the recovery factor, as presented in Table 22.

With increasing polymer molecular weight, the recovery factor further improved. When the molecular weight increased from 20 million to 25 million, the recovery factor rose by 0.74%.

(2) Effect of High-Molecular-Weight Polymer Solutions on Exploitation of Microscopic Residual Oil

The number of oil-bearing pores and throats and their respective oil saturation ratios after high-molecular-weight polymer flooding were recorded, as presented in Table 23.

The results show that, with increasing polymer molecular weight, the number of oil-bearing pores and throats slightly decreased, and the oil saturation ratio in throats reduced. This indicates that the primary effect of high-molecular-weight polymers on microscopic residual oil exploitation lies in reducing the proportion of throats with high oil saturation. As shown in the Figure 23, it illustrates the distribution of oil saturation ratios in pores and throats across different oil saturation intervals after high-concentration molecular weight polymer flooding with relative molecular weights of 2000 and 2500.

From Table 24, after high-concentration polymer flooding, as the molecular weight increased, the number of oil-bearing pores and throats in the digital pore model decreased, and the oil saturation ratio in throats decreased by 1%. The proportion of throats with high oil saturation decreased, while that of throats with low oil saturation increased, albeit with minor variations. The number of throats and pores with zero oil saturation increased. This phenomenon indicates that increasing polymer molecular weight had a limited effect on reducing the residual oil saturation, whereas high-concentration, high-molecular-weight polymers exerted a more significant impact on throats with high saturation.

### 4.3. Effect of Low-Interfacial-Tension Binary Flooding System on Exploitation of Residual Oil After Polymer Flooding

(1) Effect of Low Interfacial Tension on Recovery Factor

Building upon polymer flooding, binary flooding systems with interfacial tensions of 0.3 [mN/m] and 0.03 [mN/m] were used to study their effects on the recovery factor, as shown in Table 25.

The results demonstrate that reducing the interfacial tension can further enhance the recovery factor. When the interfacial tension was reduced from 0.3 [mN/m] to 0.03 [mN/m], the recovery factor increased by 1.12%.

(2) Effect of Low Interfacial Tension on Exploitation of Microscopic Residual Oil

The number of oil-bearing pores and throats and their respective oil saturation ratios after binary flooding were recorded, as presented in Table 26.

With decreasing interfacial tension, the number of oil-bearing pores and throats decreased, and the oil saturation ratio in throats reduced. This indicates that low interfacial tension aids in reducing oil saturation within pores, having a more pronounced effect on throats with high oil saturation. As shown in the Figure 24, it illustrates the proportion of different oil saturation distributions in pores and throats across various oil saturation intervals under interfacial tensions of 0.3 [mN/m] and 0.003 [mN/m].

From Table 26 and Table 27, it can be seen that, when the interfacial tension decreased from 10^−1^ to 10^−2^, the number of oil-bearing pores and throats both decreased, with the oil saturation ratio in throats decreasing by 1%. The proportion of throats with oil saturation between 0.8 ≤ So < 1 decreased, while the proportion with 0 < So < 0.4 increased. This indicates that reducing interfacial tension effectively lowered oil saturation within pores, with a more significant impact on pores with higher oil saturation.

## 5. Conclusions

This study, through simulating the oil displacement process under varying conditions of polymer solution concentration, relative molecular weight, and interfacial tension, delved deeply into the effects of these parameters on the distribution of microscopic residual oil and oil displacement efficiency. The results demonstrated that the judicious regulation of polymer solution concentration, relative molecular weight, and interfacial tension can optimize the distribution characteristics of microscopic residual oil and enhance oil displacement efficiency. The specific conclusions are as follows:

Effect of Polymer Concentration: With an increase in polymer concentration, the proportion of throats with high oil saturation (80~100%) decreased by over 25% after polymer flooding, whereas the proportion of other throats increased by about 50~100%. In the model, the proportion of pores with medium oil saturation (40~80%) decreased, while those with low oil saturation (0~40%) or without oil increased. Moderately increasing the polymer concentration effectively emulsified large residual oil masses, though excessively high concentrations hindered the dispersion of residual oil. As the concentration increased, clustered residual oil gradually decreased, columnar residual oil increased, while other types of residual oil showed minor changes in saturation.

Effect of Relative Molecular Weight: As the polymer relative molecular weight increased, the proportion of throats with high oil saturation decreased after polymer flooding. The proportion of throats with medium saturation initially decreased before rising, while those with low saturation first increased and then receded. The proportion of pores with high oil saturation decreased, and the increment in medium-saturation pores first grew before declining, with minor changes in low-saturation pore proportions. An increase in the relative molecular weight led to a gradual reduction in clustered residual oil and an increase in columnar residual oil, with little change in the saturation of other residual oil types.

Effect of Interfacial Tension: With the reduction in interfacial tension, the proportion of throats with high oil saturation gradually declined after polymer flooding, while the proportion of throats with medium saturation remained relatively stable, and that of throats with low saturation increased. Clustered residual oil gradually decreased, while other types of residual oil showed minimal change, and the overall recovery factor significantly improved. Reducing the interfacial tension effectively reduced residual oil in pores with high oil saturation, thereby enhancing the oil displacement performance.

Exploitation by High-Concentration, High-Molecular-Weight Polymer Solutions: After high-concentration polymer flooding, the proportion of throats with high oil saturation decreased, whereas that with low oil saturation increased. Merely increasing the polymer molecular weight had a limited effect on reducing the residual oil saturation, yet high-concentration, high-molecular-weight polymers had a more pronounced impact on throats with high saturation. Decreasing the interfacial tension effectively reduced oil saturation within pores, with greater effects observed in pores with higher oil saturation.

In summary, adjusting the polymer solution concentration, relative molecular weight, and interfacial tension plays a crucial role in optimizing the distribution of microscopic residual oil and enhancing the oil displacement efficiency. This provides scientific evidence for further improving oilfield recovery rates and achieving efficient reservoir development.

## Figures and Tables

**Figure 1 polymers-16-03246-f001:**
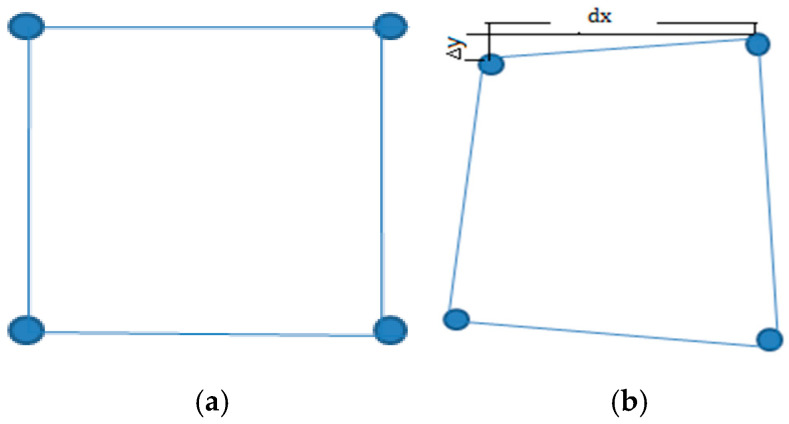
Simplified schematic of orifice–throat connection: (**a**) regular array; (**b**) irregular array.

**Figure 2 polymers-16-03246-f002:**
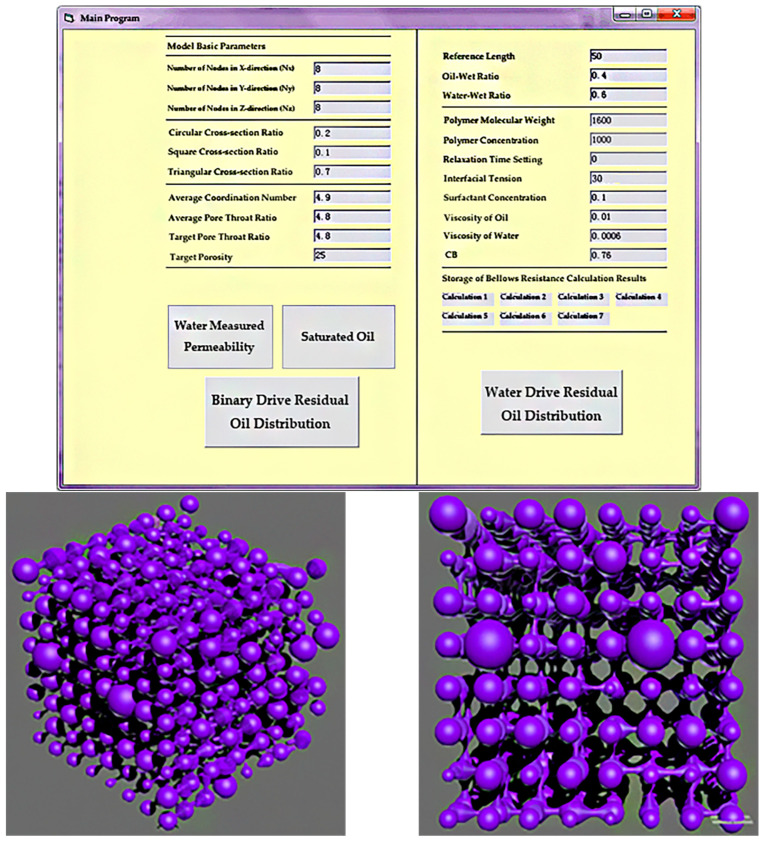
Digital pore and throat network model.

**Figure 3 polymers-16-03246-f003:**
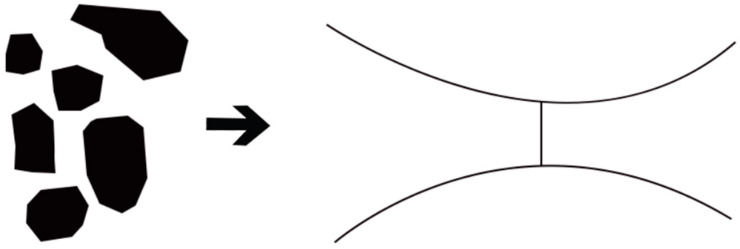
Simplified pore-throat model.

**Figure 4 polymers-16-03246-f004:**
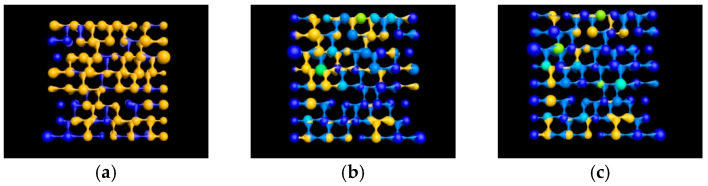
Dynamic simulation results of model flooding. (**a**) Initial model state (oil saturation: 75%); (**b**) model state after water flooding (oil saturation: 44.72%); (**c**) model state after polymer flooding (oil saturation: 31.18%).

**Figure 5 polymers-16-03246-f005:**
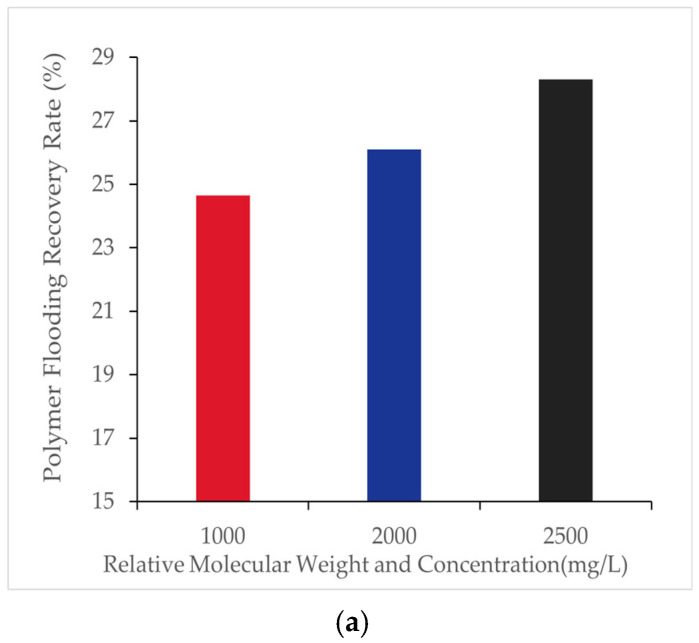
Models of different polymer concentrations.

**Figure 6 polymers-16-03246-f006:**
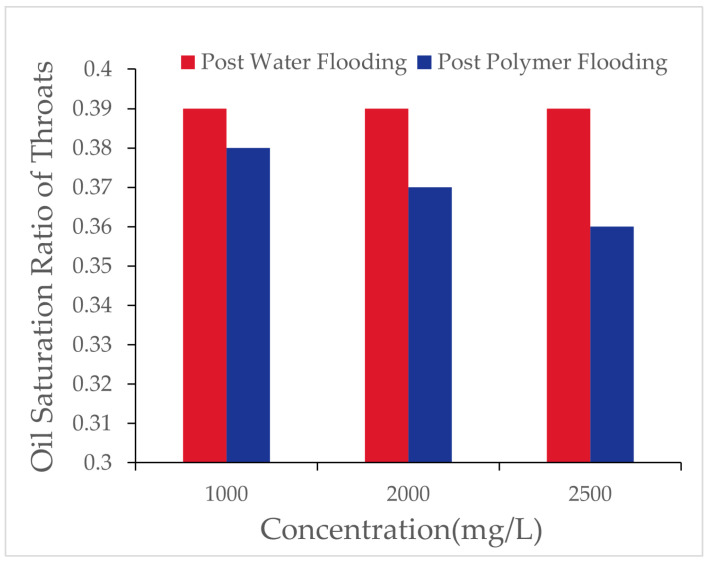
The proportion of oil content after polymer flooding with different concentrations.

**Figure 7 polymers-16-03246-f007:**
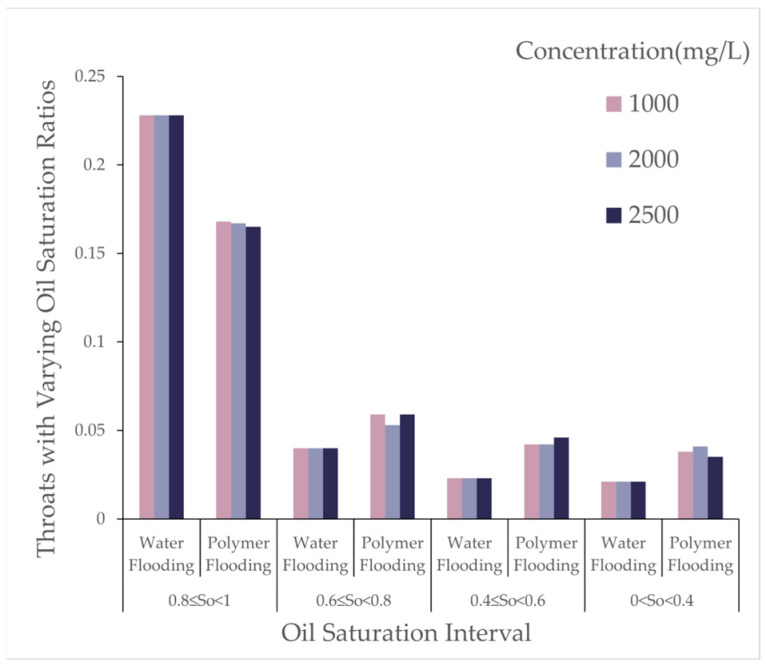
Oil saturation ratio in throats of relative molecular weight (16 million) after polymer flooding.

**Figure 8 polymers-16-03246-f008:**
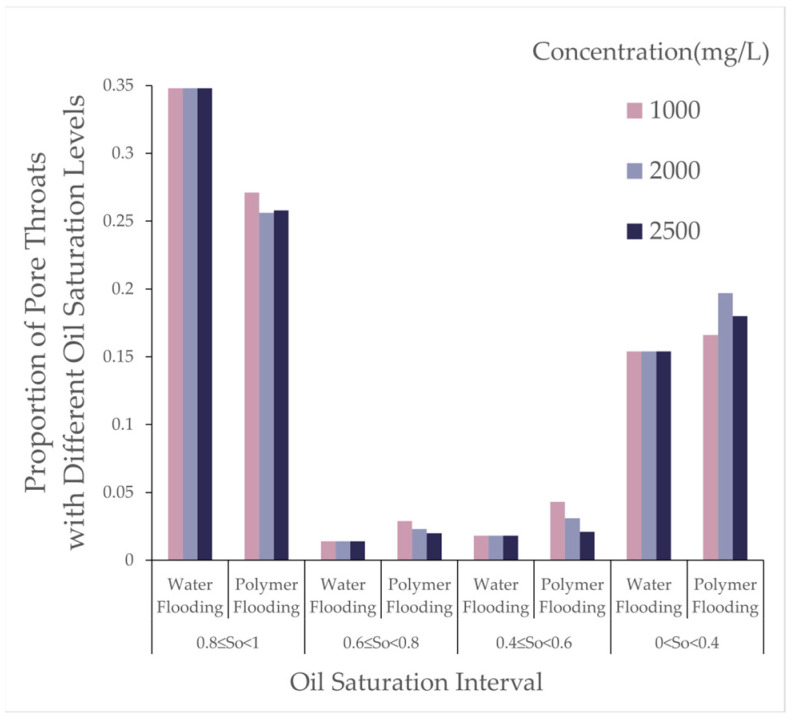
Oil saturation ratio in pores of relative molecular weight (16 million) after polymer flooding.

**Figure 9 polymers-16-03246-f009:**
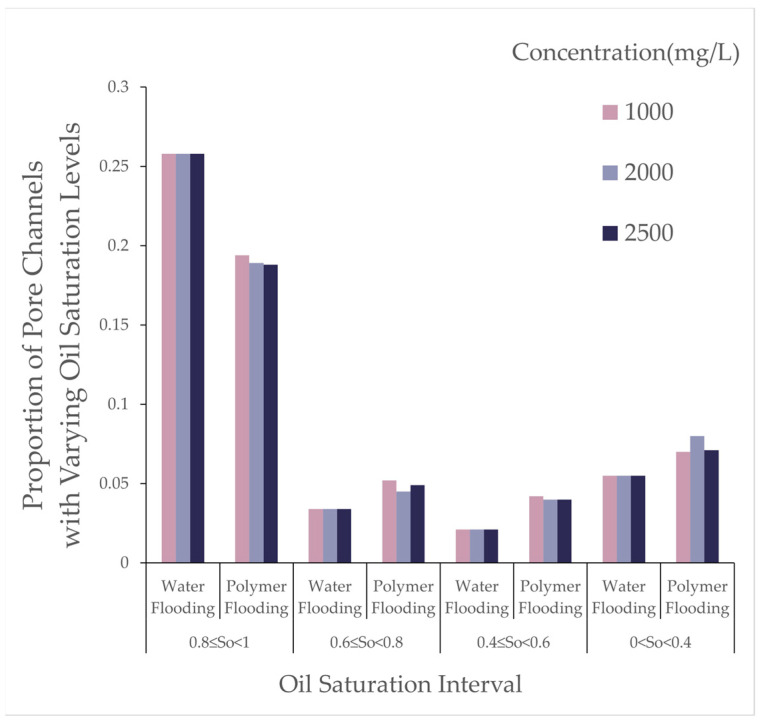
Oil saturation ratio in pore throat after polymer flooding with relative molecular weight (16 million).

**Figure 10 polymers-16-03246-f010:**
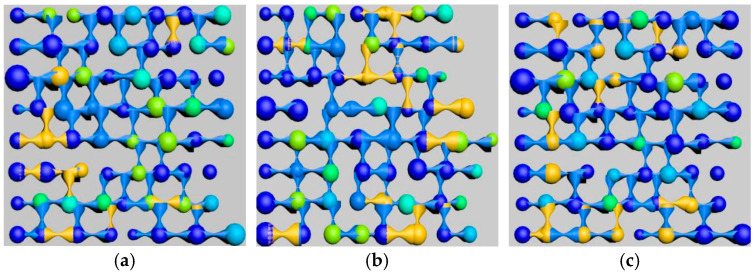
Effects of different polymer concentrations on residual oil distribution: (**a**) 1000 [mg/L]; (**b**) 2000 [mg/L]; (**c**) 2500 [mg/L].

**Figure 11 polymers-16-03246-f011:**
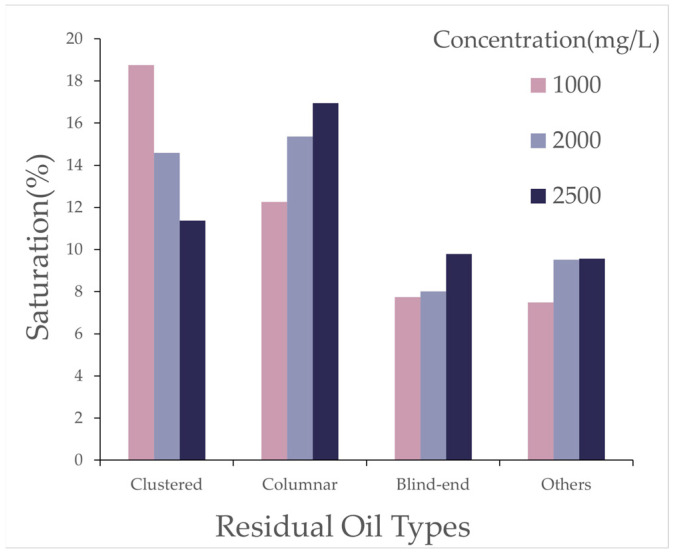
Effects of different polymer concentrations on residual oil types.

**Figure 12 polymers-16-03246-f012:**
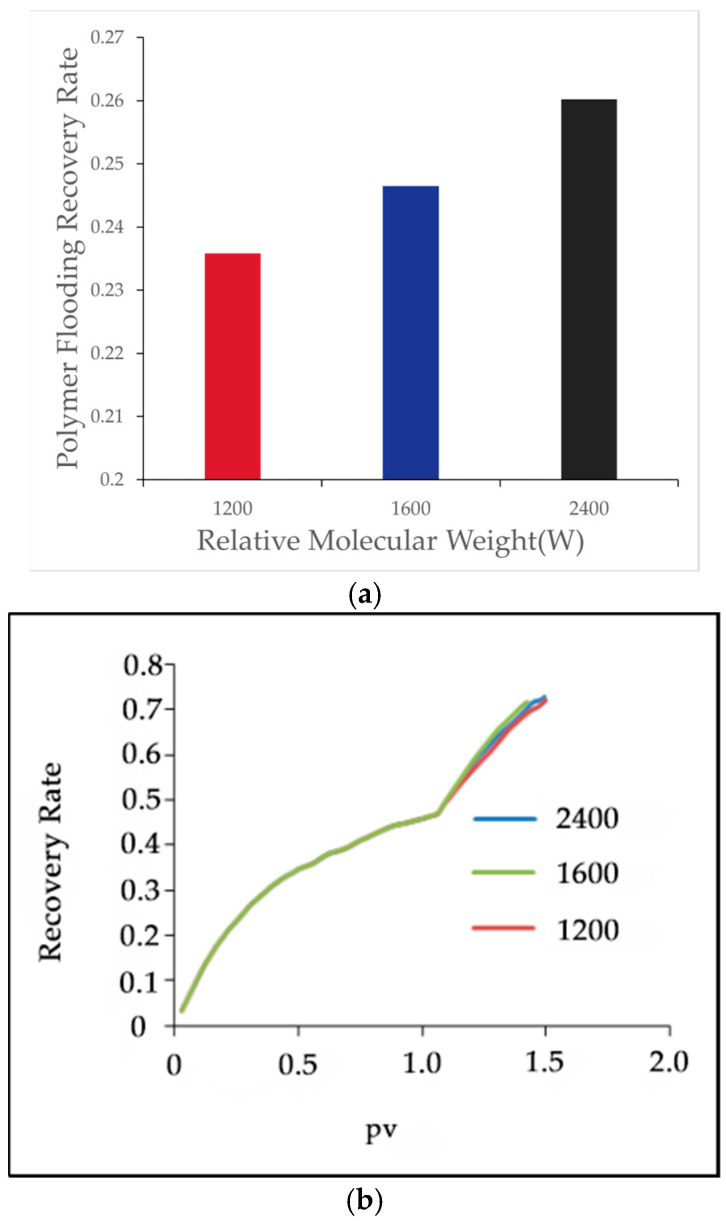
Models of different polymers’ relative molecular masses.

**Figure 13 polymers-16-03246-f013:**
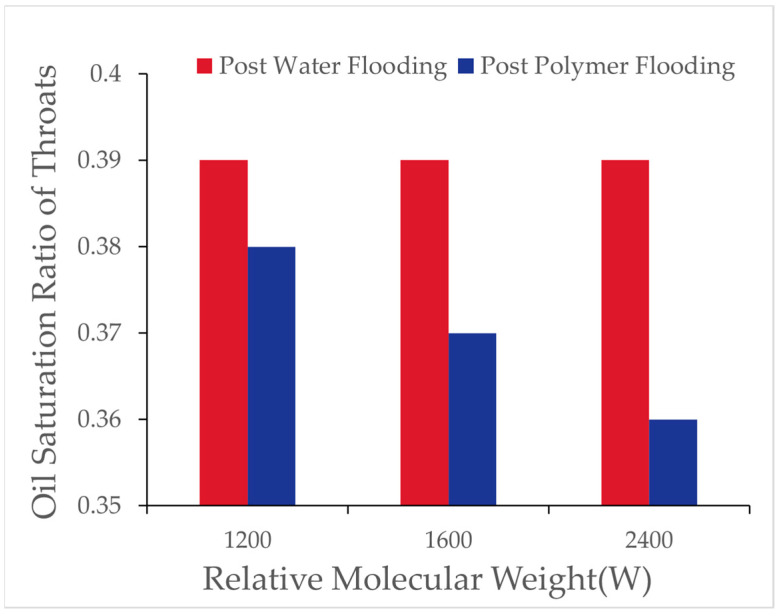
Oil saturation ratio in pore throat after polymer flooding.

**Figure 14 polymers-16-03246-f014:**
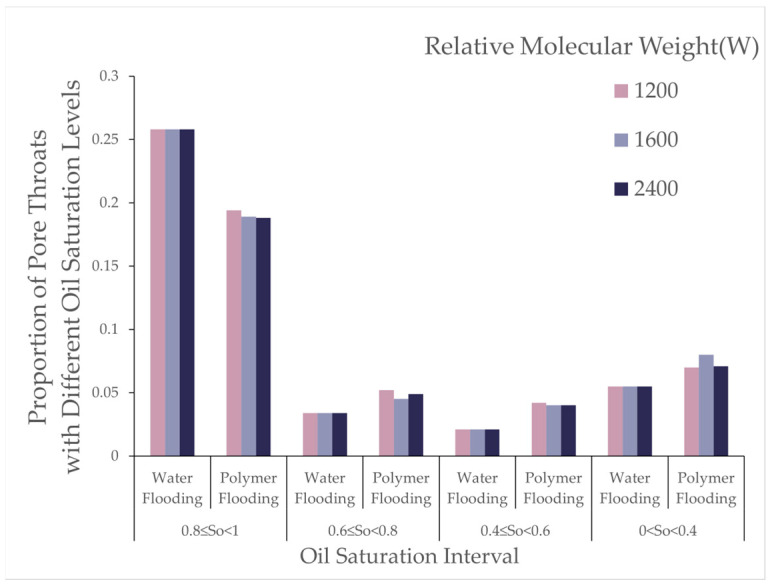
Oil saturation ratio of pore throats after polymer flooding (1000 [mg/L]).

**Figure 15 polymers-16-03246-f015:**
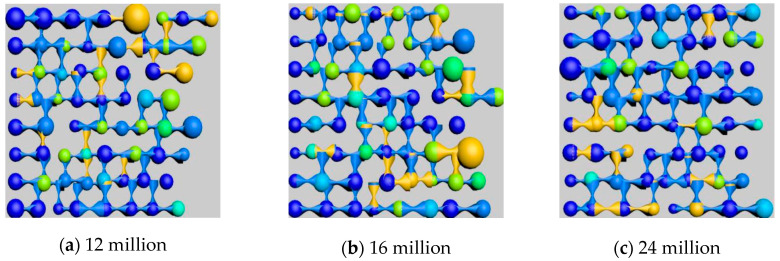
Effects of different polymer molecular weights on residual oil distribution.

**Figure 16 polymers-16-03246-f016:**
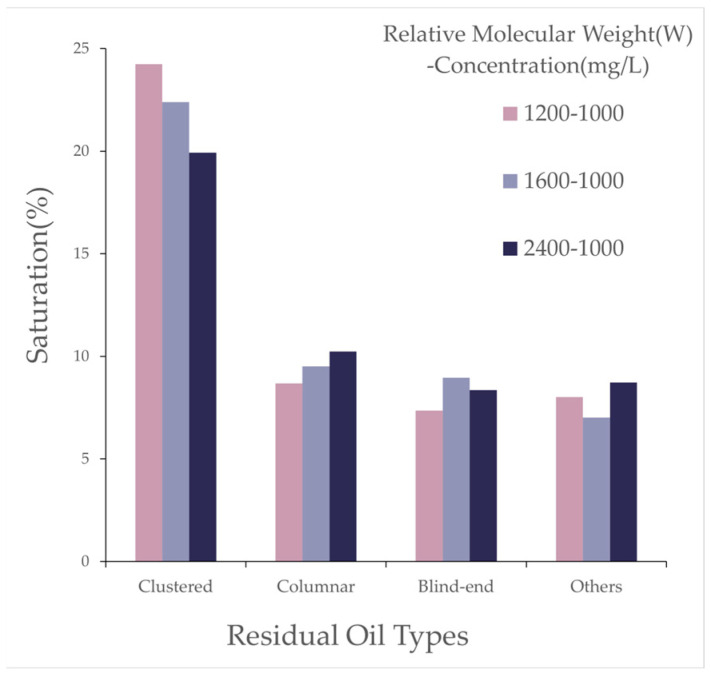
Effects of different polymer molecular weights on residual oil types.

**Figure 17 polymers-16-03246-f017:**
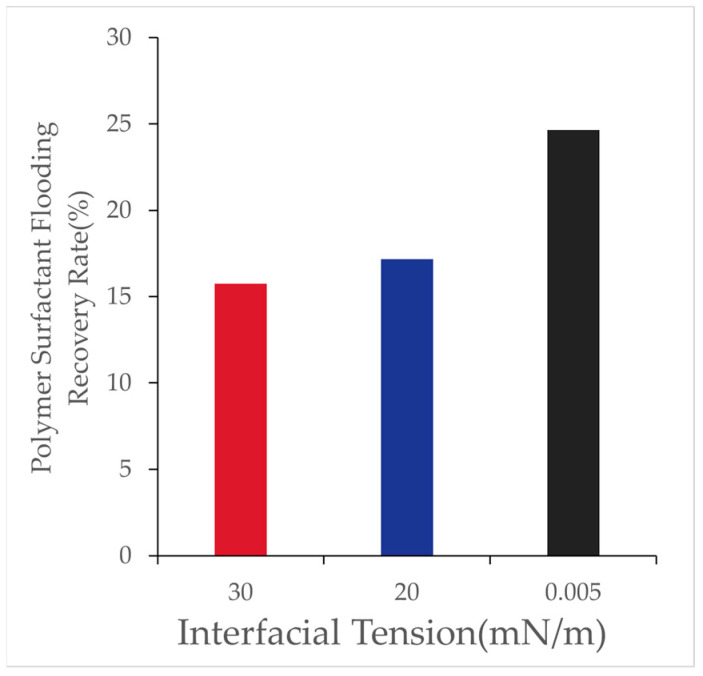
Influence of interfacial tension on recovery factor in polymer–surfactant binary flooding.

**Figure 18 polymers-16-03246-f018:**
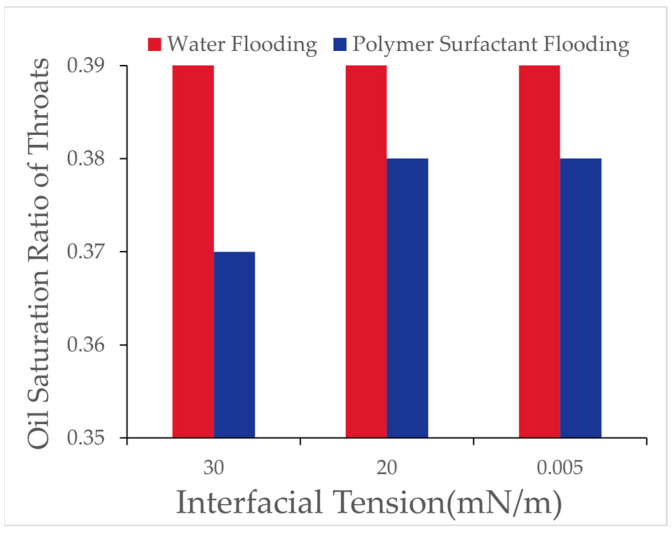
Oil saturation ratio in pore throat after polymer–surfactant binary flooding.

**Figure 19 polymers-16-03246-f019:**
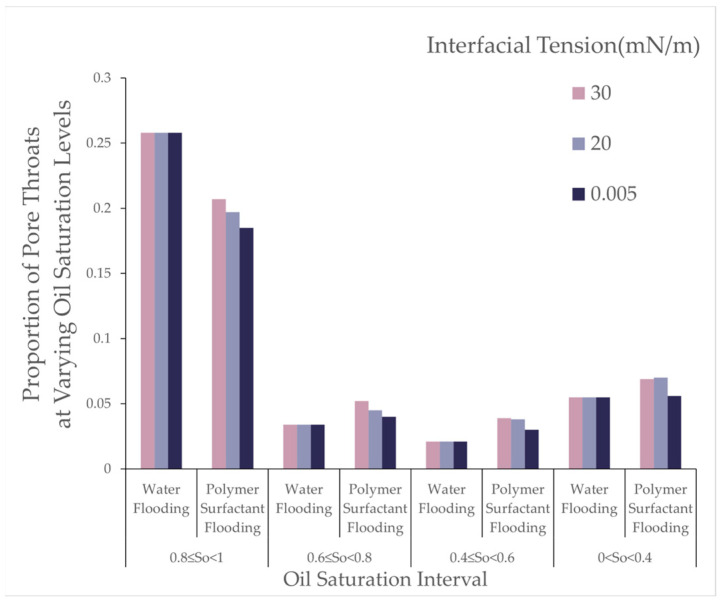
Oil saturation ratio in pore throat.

**Figure 20 polymers-16-03246-f020:**
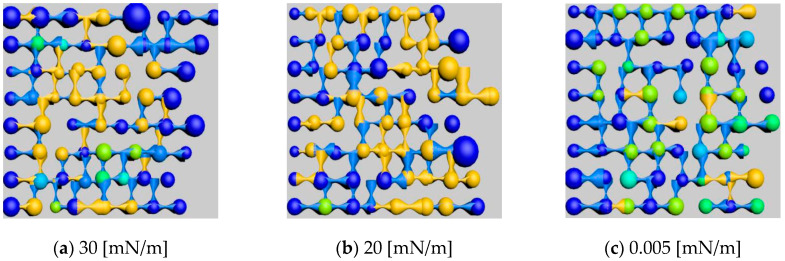
Influence of different interfacial tensions on residual oil distribution.

**Figure 21 polymers-16-03246-f021:**
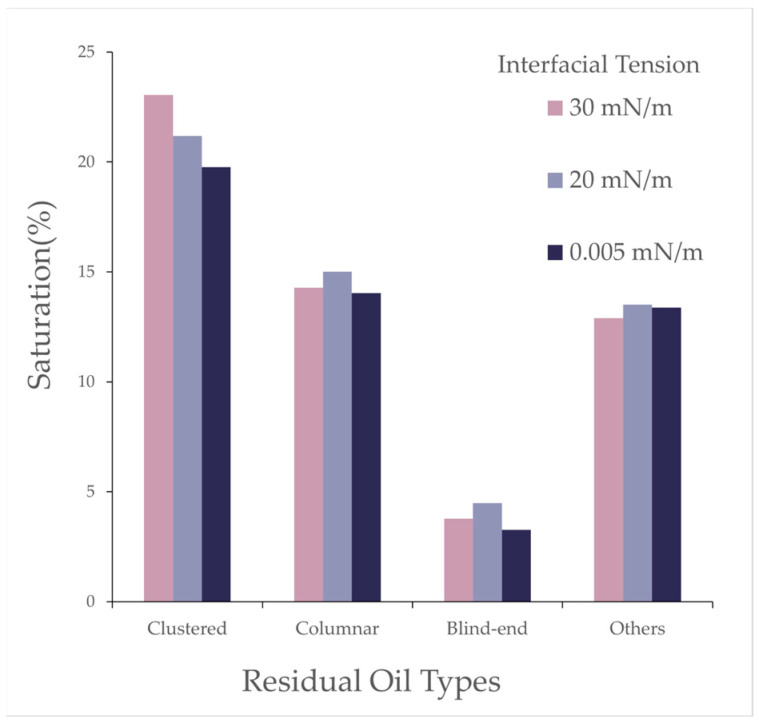
Effect of different interfacial tensions on residual oil types.

**Figure 22 polymers-16-03246-f022:**
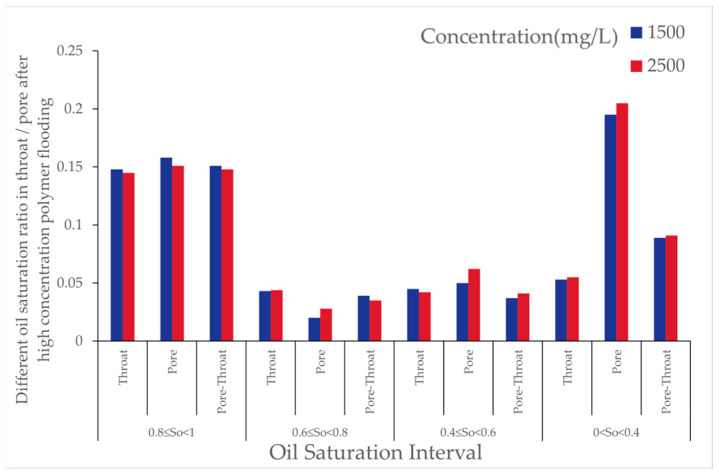
Different oil saturation ratios in pore throat after high-concentration polymer flooding (molecular weight 20 million).

**Figure 23 polymers-16-03246-f023:**
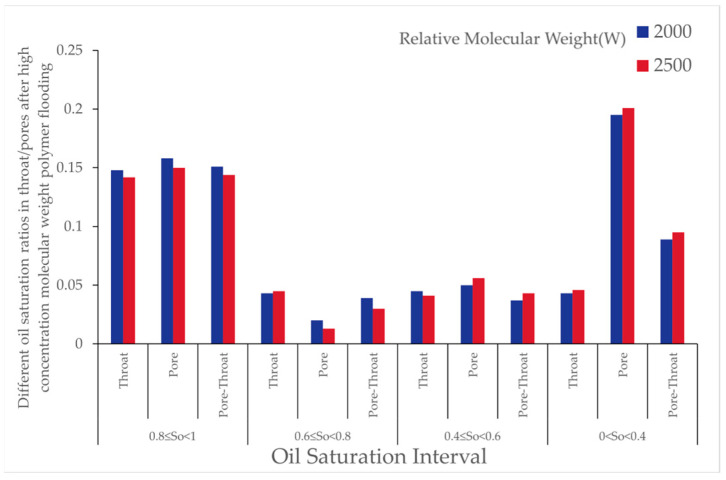
Different oil saturation ratios in pore throat after high-molecular-weight polymer flooding (concentration 1500 [mg/L]).

**Figure 24 polymers-16-03246-f024:**
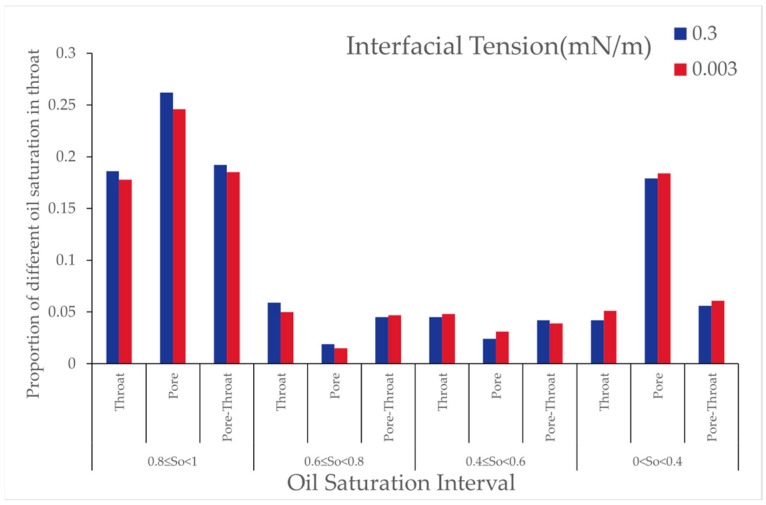
Different oil saturation ratios in pore throat.

**Table 1 polymers-16-03246-t001:** Basic parameters of network model.

Core Number Model Parameters	#1	#2	#3	#4	#5
Porosity	25.1	24.6	24.1	26.3	27.6
water measurement permeability	370	39	430	568	623
the length of the larynx	21~187	3~65	8~155	15~167	16~168
throat radius	1~20	1~16	1~18	1~22	1~25
mean coordination number	4.2	4.3	4.6	4.1	4.2
average pore-throat ratio	5.4	5.1	4.7	4.3	4.2

**Table 2 polymers-16-03246-t002:** Effect of polymer concentration on oil displacement efficiency.

Concentration (mg/L)	1000	2000	2500
Water Flooding Recovery Rate (%)	47.07	47.07	47.07
Polymer Flooding Recovery Rate (%)	24.65	26.10	28.30
Total Recovery Rate (%)	71.72	72.06	72.37

**Table 3 polymers-16-03246-t003:** Oil saturation ratio in pore throat after polymer flooding with different concentrations.

Concentration (mg/L)		1000	2000	2500
Total	Total Throat Count	993	993	993
	Total Pore Count	512	512	512
After Water Flooding	Throat with Oil Count	310	310	310
	Pore with Oil Count	273	273	273
	Total Oil-Bearing Count	583	583	583
	Throat Oil-Bearing Ratio	0.39	0.39	0.39
After Polymer Flooding	Throat with Oil Count	305	301	302
	Pore with Oil Count	261	260	245
	Total Oil-Bearing Count	566	510	460
	Throat Oil-Bearing Ratio	0.38	0.37	0.36

**Table 4 polymers-16-03246-t004:** Oil saturation ratio in throats of relative molecular weight (16 million) after polymer flooding.

Oil Saturation	0.8 ≤ So < 1		0.6 ≤ So < 0.8		0.4 ≤ So < 0.6		0 < So < 0.4		0	
Concentration	Water Flooding	Polymer Flooding	Water Flooding	Polymer Flooding	Water Flooding	Polymer Flooding	Water Flooding	Polymer Flooding	Water Flooding	Polymer Flooding
1000	0.228	0.168	0.040	0.059	0.023	0.042	0.021	0.038	0.688	0.693
2000	0.228	0.167	0.040	0.053	0.023	0.042	0.021	0.041	0.688	0.697
2500	0.228	0.165	0.040	0.059	0.023	0.046	0.021	0.035	0.688	0.695

**Table 5 polymers-16-03246-t005:** Oil saturation ratio in pores of relative molecular weight (16 million) after polymer flooding.

Oil Saturation	0.8 ≤ So < 1		0.6 ≤ So < 0.8		0.4 ≤ So < 0.6		0 < So < 0.4		0	
Concentration	Water Flooding	Polymer Flooding	Water Flooding	Polymer Flooding	Water Flooding	Polymer Flooding	Water Flooding	Polymer Flooding	Water Flooding	Polymer Flooding
1000	0.348	0.271	0.014	0.029	0.018	0.043	0.154	0.166	0.467	0.490
2000	0.348	0.256	0.014	0.023	0.018	0.031	0.154	0.197	0.467	0.492
2500	0.348	0.258	0.014	0.020	0.018	0.021	0.154	0.180	0.467	0.521

**Table 6 polymers-16-03246-t006:** Oil saturation ratio in pore throat after polymer flooding with relative molecular weight (16 million).

Oil Saturation	0.8 ≤ So < 1		0.6 ≤ So < 0.8		0.4 ≤ So < 0.6		0 < So < 0.4		0	
Concentration	Water Flooding	Polymer Flooding	Water Flooding	Polymer Flooding	Water Flooding	Polymer Flooding	Water Flooding	Polymer Flooding	Water Flooding	Polymer Flooding
1000	0.258	0.194	0.034	0.052	0.021	0.042	0.055	0.070	0.632	0.642
2000	0.258	0.189	0.034	0.045	0.021	0.040	0.055	0.080	0.632	0.646
2500	0.258	0.188	0.034	0.049	0.021	0.040	0.055	0.071	0.632	0.651

**Table 7 polymers-16-03246-t007:** Effects of different polymer concentrations on residual oil types.

Type Saturation	Saturation (%)		
	1000 [mg/L]	2000 [mg/L]	2500 [mg/L]
Clustered	18.76	14.58	11.37
Columnar	12.26	15.36	16.95
Blind-End	7.74	8.01	9.78
Others	7.49	9.52	9.56

**Table 8 polymers-16-03246-t008:** Effect of polymer molecular weight on oil displacement efficiency during polymer flooding.

Relative Molecular Weight (in 10,000 s)	1200	1600	2400
Water Flooding Extraction Rate (%)	47.07	47.07	47.07
Polymer Flooding Extraction Rate (%)	23.58	24.65	26.02
Total Extraction Rate (%)	70.65	71.72	73.09

**Table 9 polymers-16-03246-t009:** Oil saturation ratio in pore throat after polymer flooding with different molecular weights.

Relative Molecular Weight (in 10,000 s)		1200	1600	2400
Total	Total Throat Count	993	993	993
	Total Pore Count	512	512	512
After Water Flooding	Throat with Oil Count	310	310	310
	Pore with Oil Count	273	273	273
	Total Oil-Bearing Count	583	583	583
	Throat Oil-Bearing Ratio	0.39	0.39	0.39
After Polymer Flooding	Throat with Oil Count	304	301	302
	Pore with Oil Count	261	260	257
	Total Oil-Bearing Count	565	561	559
	Throat Oil-Bearing Ratio	0.38	0.37	0.36

**Table 10 polymers-16-03246-t010:** Oil saturation ratio of pore throats after polymer flooding (1000 [mg/L]).

Oil Saturation	0.8 ≤ So < 1		0.6 ≤ So < 0.8		0.4 ≤ So < 0.6		0 < So < 0.4		0	
Relative Molecular Weight (in 10,000 s)	Water Flooding	Polymer Flooding	Water Flooding	Polymer Flooding	Water Flooding	Polymer Flooding	Water Flooding	Polymer Flooding	Water Flooding	Polymer Flooding
1200	0.258	0.194	0.034	0.052	0.021	0.042	0.055	0.070	0.632	0.642
1600	0.258	0.189	0.034	0.045	0.021	0.040	0.055	0.080	0.632	0.646
2400	0.258	0.188	0.034	0.049	0.021	0.040	0.055	0.071	0.632	0.651

**Table 11 polymers-16-03246-t011:** Effects of different polymer molecular weights on residual oil types.

Type	Saturation (%)		
	1200–1000 [mg/L]	1600–1000 [mg/L]	2400–1000 [mg/L]
Clustered	24.23	22.39	19.92
Columnar	8.67	9.51	10.24
Blind End	7.35	8.95	8.35
Others	8.01	7.02	8.72

**Table 12 polymers-16-03246-t012:** Effect of interfacial tension on recovery factor.

Interfacial Tension (mN/m)	30	20	0.005
Water Flooding Recovery Rate (%)	47.07	47.07	47.07
Binary Flooding Recovery Rate (%)	15.76	17.18	24.65

**Table 13 polymers-16-03246-t013:** Oil saturation ratio in pore throat after polymer–surfactant binary flooding.

Interfacial Tension (mN/m)		30	20	0.005
Total	Total Throat Count	993	993	993
	Total Pore Count	512	512	512
After Water Flooding	Throat with Oil Count	310	310	310
	Pore with Oil Count	273	273	273
	Total Oil-Bearing Count	583	583	583
	Throat Oil-Bearing Ratio	0.39	0.39	0.39
After Binary Flooding	Throat with Oil Count	301	295	280
	Pore with Oil Count	261	246	237
	Total Oil-Bearing Count	562	570	577
	Throat Oil-Bearing Ratio	0.37	0.38	0.38

**Table 14 polymers-16-03246-t014:** Oil saturation ratio in pore throat.

Oil Saturation	0.8 ≤ So < 1		0.6 ≤ So < 0.8		0.4 ≤ So < 0.6		0 < So < 0.4		0	
Interfacial Tension (mN/m)	Water Flooding	Polymer Surfactant Flooding	Water Flooding	Polymer Surfactant Flooding	Water Flooding	Polymer Surfactant Flooding	Water Flooding	Polymer Surfactant Flooding	Water Flooding	Polymer Surfactant Flooding
30	0.258	0.207	0.034	0.052	0.021	0.039	0.055	0.069	0.632	0.633
20	0.258	0.197	0.034	0.045	0.021	0.038	0.055	0.070	0.632	0.649
0.005	0.258	0.185	0.034	0.040	0.021	0.030	0.055	0.056	0.632	0.625

**Table 15 polymers-16-03246-t015:** Effect of different interfacial tensions on residual oil types.

Type	Saturation (%)		
	30 [mN/m]	20 [mN/m]	0.005 [mN/m]
Clustered	23.05	21.18	19.76
Columnar	14.29	15.01	14.03
Blind End	3.77	4.49	3.26
Others	12.89	13.52	13.37

**Table 19 polymers-16-03246-t019:** Effect of high-concentration polymer solutions on recovery factor after polymer flooding.

Concentration (mg/L)	1500	2500
Water Flooding Recovery Rate (%)	47.07	47.07
Polymer Flooding Recovery Rate (%)	24.65	24.65
High-Concentration Polymer Flooding Recovery Rate (%)	3.05	4.57
Total Recovery Rate (%)	74.77	76.29

**Table 20 polymers-16-03246-t020:** Oil saturation ratio in pore throat after high-concentration polymer flooding.

Concentration (mg/L)		1500	2500
Total	Total Throat Count	993	993
	Total Pore Count	512	512
After High-Concentration Polymer Flooding	Throat with Oil Count	287	284
	Pore with Oil Count	255	250
	Total Oil-Bearing Count	542	534
	Throat Oil-Bearing Ratio	0.36	0.35

**Table 21 polymers-16-03246-t021:** Different oil saturation ratios in pore throat after high-concentration polymer flooding (molecular weight 20 million).

Type	Concentration	Oil Saturation				
		0.8 ≤ So < 1	0.6 ≤ So < 0.8	0.4 ≤ So < 0.6	0 < So < 0.4	0
Throat	1500	0.148	0.043	0.045	0.053	0.711
	2500	0.145	0.044	0.042	0.055	0.714
Pore	1500	0.158	0.02	0.05	0.195	0.577
	2500	0.151	0.028	0.062	0.205	0.554
Pore + Throat	1500	0.151	0.039	0.037	0.089	0.684
	2500	0.148	0.035	0.041	0.091	0.685

**Table 22 polymers-16-03246-t022:** Effect of high-molecular-weight polymer solutions on recovery factor after polymer flooding.

Relative Molecular Weight (in 10,000 s)	2000	2500
Water Flooding Recovery Rate (%)	47.07	47.07
Polymer Flooding Recovery Rate (%)	24.65	24.65
High-Concentration Polymer Flooding Recovery Rate (%)	3.05	3.79
Total Recovery Rate (%)	74.77	75.51

**Table 23 polymers-16-03246-t023:** Oil saturation ratio in pore throat after high-molecular-weight polymer flooding.

Relative Molecular Weight (in 10,000 s)		2000	2500
Total	Total Throat Count	993	993
	Total Pore Count	512	512
After High-Concentration Polymer Flooding	Throat with Oil Count	277	272
	Pore with Oil Count	255	254
	Total Oil-Bearing Count	532	526
	Throat Oil-Bearing Ratio	0.35	0.34

**Table 24 polymers-16-03246-t024:** Different oil saturation ratios in pore throat after high-molecular-weight polymer flooding (concentration 1500 [mg/L]).

Type	Molecular Weight (in 10,000 s)	Oil Saturation				
		0.8 ≤ So < 1	0.6 ≤ So < 0.8	0.4 ≤ So < 0.6	0 < So < 0.4	0
Throat	2000	0.148	0.043	0.045	0.043	0.721
	2500	0.142	0.045	0.041	0.046	0.726
Pore	2000	0.158	0.02	0.05	0.195	0.577
	2500	0.15	0.013	0.056	0.201	0.58
Pore + Throat	2000	0.151	0.039	0.037	0.089	0.684
	2500	0.144	0.03	0.043	0.095	0.688

**Table 25 polymers-16-03246-t025:** Effect of low interfacial tension on recovery factor.

Interfacial Tension (mN/m)	0.3	0.03
Water Flooding Recovery Rate (%)	47.07	47.07
Polymer Flooding Recovery Rate (%)	24.65	24.65
Polymer Flooding Recovery Rate (%)	2.13	3.25
Total Recovery Rate (%)	73.85	74.97

**Table 26 polymers-16-03246-t026:** Oil saturation ratio in pore throat after binary flooding.

Interfacial Tension (mN/m)		0.3	0.003
Total	Total Throat Count	993	993
	Total Pore Count	512	512
After Polymer Flooding	Throat with Oil Count	330	325
	Pore with Oil Count	247	235
	Total Oil-Bearing Count	577	560
	Throat Oil-Bearing Ratio	0.38	0.37

**Table 27 polymers-16-03246-t027:** Different oil saturation ratios in pore throat.

Type	Interfacial Tension (mN/m)	Oil Saturation				
		0.8 ≤ So < 1	0.6 ≤ So < 0.8	0.4 ≤ So < 0.6	0 < So < 0.4	0
Throat	0.3	0.186	0.059	0.045	0.042	0.668
	0.003	0.178	0.05	0.048	0.051	0.673
Pore	0.3	0.262	0.019	0.024	0.179	0.516
	0.003	0.246	0.015	0.031	0.184	0.524
Pore + Throat	0.3	0.192	0.045	0.042	0.056	0.665
	0.003	0.185	0.047	0.039	0.061	0.668

## Data Availability

The original contributions presented in the study are included in the article, further inquiries can be directed to the corresponding author.

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
