# Peer review of "Study on the Microscopic Distribution Pattern of Residual Oil and Exploitation Methods Based on a Digital Pore Network Model"

_polymers, 2024, doi:10.3390/polym16233246_

Round 1
Reviewer 1 Report
Comments and Suggestions for Authors
This paper uses digital pore network model to study the microscopic distribution pattern of residual oil and exploitation methods. I recommend that this work be published in the Polymers after full text and detailed formatting review and revision.
1. In a paper, units should be written using "[]".
2. What is the reason for half of the content in Table 3-4 being bolded?
3. What are the specific reasons why different polymer molecular weights affect residual oil types?
4. Please pay attention to the formatting of the manuscript, especially the references. The author should review it carefully.
Reviewer 2 Report
Comments and Suggestions for Authors
The reviewed paper: "Study on the Microscopic Distribution Pattern of Residual Oil and Exploitation Methods Based on a Digital Pore Network Mode" presents the theoretical studies on the influence of various parameters on the oil displacement process from a polymer. Unfortunately, the presented research is only theoretical research and computer simulations. In the presented article they have no confirmation in experimental studies. Therefore, their presentation raises great concerns as to whether the theoretical experiment can be applied in practice in polymer flooding technology.
Therefore, in my opinion, the article in its current form is not suitable for publication and should be supplemented with experimental data. The theoretical data obtained should be compared with experimental data and it should be clarified experimentally whether the developed model is suitable for these polymers and whether it is a universal model (suitable for all polymers).
In addition, in this paper there are too many theoretical foundations described that are well known.
There is no part characterizing the structure of the polymer used for simulation.
Round 2
Reviewer 2 Report
Comments and Suggestions for Authors
This paper was improved according to my comments.
It is suitable for publication in its current form.